# Backjump-on-Graph: Empowering Large Language Models with Reinforced Retrospective Exploration for Agentic Knowledge Graph Reasoning

Yunqi Zhang [1]  Shiqi Yan [1,2]  Zhenzhao Yuan [2]  Wenrui Liang [1,2]  Yangming Liu [2]  Zhixiao Qi [1,2]  Tianyi Zhang [3]
Shijie Zhang [3]  Wei-Qiang Zhang [1,2]  Yongfeng Huang [1,2]  Haixin Duan [1,4]  Shuai Chen [3]  Yubo Chen [1]

## Abstract

Grounding Large Language Models (LLMs) in Knowledge Graphs (KGs) has shown significant promise for complex Question Answering (QA) tasks. Since LLMs' limited context window cannot accommodate the sheer volume of large-scale KGs, existing work usually utilizes agents to reason on real-world KGs, which follows reasoning paths derived from the queries step by step. However, the mismatch between query-derived paths and the KG's structure, stemming from users' lack of schema knowledge, usually leads the agents into dead ends. To address this problem, in this paper, we propose Backjump-on-Graph (BoG), a novel framework that empowers LLMs to retrospectively explore alternative reasoning paths at dead ends. We first propose to formalize each reasoning step with four atomic operations to create a structural scaffold that allows LLMs to revert to historical status. Next, we fine-tune the LLM with synthetic data containing the above atomic operations to instill basic backjump abilities. Finally, we leverage reinforcement learning and propose a hybrid reward function, which penalizes redundant transitions and promotes correct answers, to optimize the timing and landing nodes of backjumping. Extensive experiments on several KGQA benchmark datasets demonstrate the effectiveness of our BoG method. Code is available at https://github.com/zhangSchnee/BoG.

## 1. Introduction

Large Language Models (LLMs) have demonstrated remarkable reasoning capabilities in various natural language processing tasks (Brown et al., 2020; Qiao et al., 2023; Dubey et al., 2024). However, they often suffer from hallucinations and a lack of verifiable evidence in complex Question Answering (QA) tasks (Mallen et al., 2023; Pan et al., 2024). To mitigate this limitation, grounding LLMs on Knowledge Graphs (KGs) has emerged as a critical solution (Radhakrishnan et al., 2023; Sui et al., 2025). By anchoring LLMs in structured knowledge, KG-enhanced methods not only alleviate hallucinations but also provide traceable evidence, significantly improving the reliability of answer derivation (Wang et al., 2023a; Luo et al., 2024).

While incorporating external facts from KGs helps augment LLMs (Zhang et al., 2022; Luo et al., 2024), the limited context window necessitates aggressive pruning of large subgraphs. This inevitably discards rich structural information and restricts the LLM's effective exploration space (Sun et al., 2023; Xie et al., 2024). To address this limitation, recent research has increasingly shifted toward agent-based paradigms (Zhu et al., 2025; Jiang et al., 2025), which leverage LLMs to reason over KGs in an iterative and decision-driven manner. Instead of retrieving a large subgraph at once, these methods follow query-induced reasoning paths and traverse the KG step by step, exploring only a small set of question-relevant neighbors at each step.

However, a significant gap exists between natural language expressions and the formal topology of KGs. Since users are generally unfamiliar with the underlying schema of a KG, the paths envisioned from the queries by the agents often diverge from the ground-truth connectivity in the KG, causing the reasoning process to stall or deviate into incorrect sub-graphs. For example, as shown in Figure 1, a conventional agent might greedily follow the "*same_universe*" and "*director*" relations based on the local semantic proximity of the question. This choice finally leads to a dead-end node "*Paul W.S. Anderson*". Despite this failure, we observe that a viable path exists at the intermediate node "*Prometheus*" via the "*prequel*" relation; although different from "*same_universe*", it remains semantically relevant and

[1]Zhongguancun Laboratory [2]Department of Electronic Engineering, Tsinghua University [3]Ant International, Ant Group [4]Institute for Network Sciences and Cyberspace, Tsinghua University. Correspondence to: Yubo Chen <ybch14@gmail.com>.

*Proceedings of the 43rd International Conference on Machine Learning*, Seoul, South Korea. PMLR 306, 2026. Copyright 2026 by the author(s).

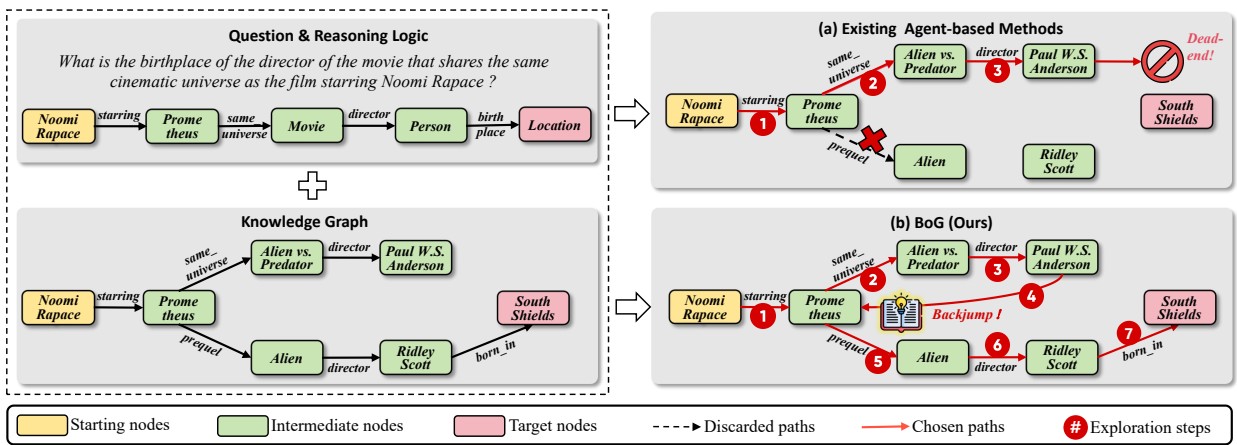

*Figure 1.* Comparison between existing agent-based KG navigation and our backjumping method when encountering dead ends.

leads to the ground truth. This case suggests that if the agent could perform a backjump directly to this pivotal junction and re-explore alternative branches, the correct answer would remain reachable. Such an observation motivates the need for a backjumping mechanism, enabling LLM agents to escape reasoning dead-ends by non-locally reverting to promising decision points.

In this paper, we propose Backjump-on-Graph (BoG), a novel reasoning framework that enables LLMs to automatically recover from topological traps through backjumping. First, we propose to formalize each reasoning step into four atomic operations—*Forward*, *Backjump*, *Yield*, and *Halt*—thereby constructing a structural scaffold that explicitly exposes historical states and allows the model to revert to earlier decision points. Next, we synthesize a dataset incorporating these four atomic operations and fine-tune the LLM, effectively endowing it with preliminary backjumping capabilities. Finally, we adopt reinforcement learning and propose a hybrid reward function, which penalizes redundant transitions and promotes correct answers, to optimize the backjumping strategy, including both the timing and the landing node of each backjump operation. We conduct experiments on several KGQA benchmark datasets, and the results prove the effectiveness of our method.

## 2. Related Works

Early work of Knowledge Graph Question Answering (KGQA) methods focused on symbolic reasoning to map natural language questions into logical queries on graph data (Sun et al., 2018; 2020; Li & Ji, 2022). Although these methods offered strong interpretability, they suffered from non-executable queries caused by structural variations between various questions and real KGs (Yu et al., 2023).

Subsequent work shifted to retrieval-augmented KGQA paradigms, which retrieved relevant triples from KGs to identify candidate answers (Jiang et al., 2023; Ma et al.,

2024; He et al., 2024). For example, RoG (Luo et al., 2024) integrated explicit planning with pruned relational triples to guide multi-hop reasoning. GNN-RAG (Mavromatis & Karypis, 2025) adopted graph neural networks (Kipf, 2016) to model distant dependencies within the retrieved subgraphs. However, they were constrained by the limited scope of subgraph retrieval, failing to utilize the full structural dependencies of KGs (Xie et al., 2024).

To overcome these limitations, agent-based methods iteratively traversed KGs by selecting neighboring nodes according to the given questions (Sun et al., 2023; Chen et al., 2024). For example, ToG (Sun et al., 2023) constructed different pre-defined prompts to guide LLMs in sequentially expanding entities and relations. KG-Agent (Jiang et al., 2025) introduced three types of tools for graph search, dynamically selected based on the current reasoning state. KBQA-o1 (Luo et al., 2025a) further introduced Monte Carlo Tree Search (Browne et al., 2012) to generate exploration trajectories for training, but it remained constrained to a single-path, non-revisable search process in inference. Although PoG (Chen et al., 2024) incorporated backtrack attempts to recover from dead ends, its myopic scope prevents it from performing the non-local jumps necessary to escape deep-seated reasoning traps.

Different from existing approaches, our method distinguishes itself by enabling multi-level backjumping, which allows the agent to revert to any previous historical states and explore alternative trajectories when encountering dead-ends. Furthermore, to guide this flexible exploration, we propose a specific reward function within a reinforcement learning framework, effectively optimizing the backjumping policy and enhancing overall reasoning efficiency.

## 3. Methodology

In this section, we present the technical details of our Backjump-on-Graph (BoG) framework, as illustrated in Fig-

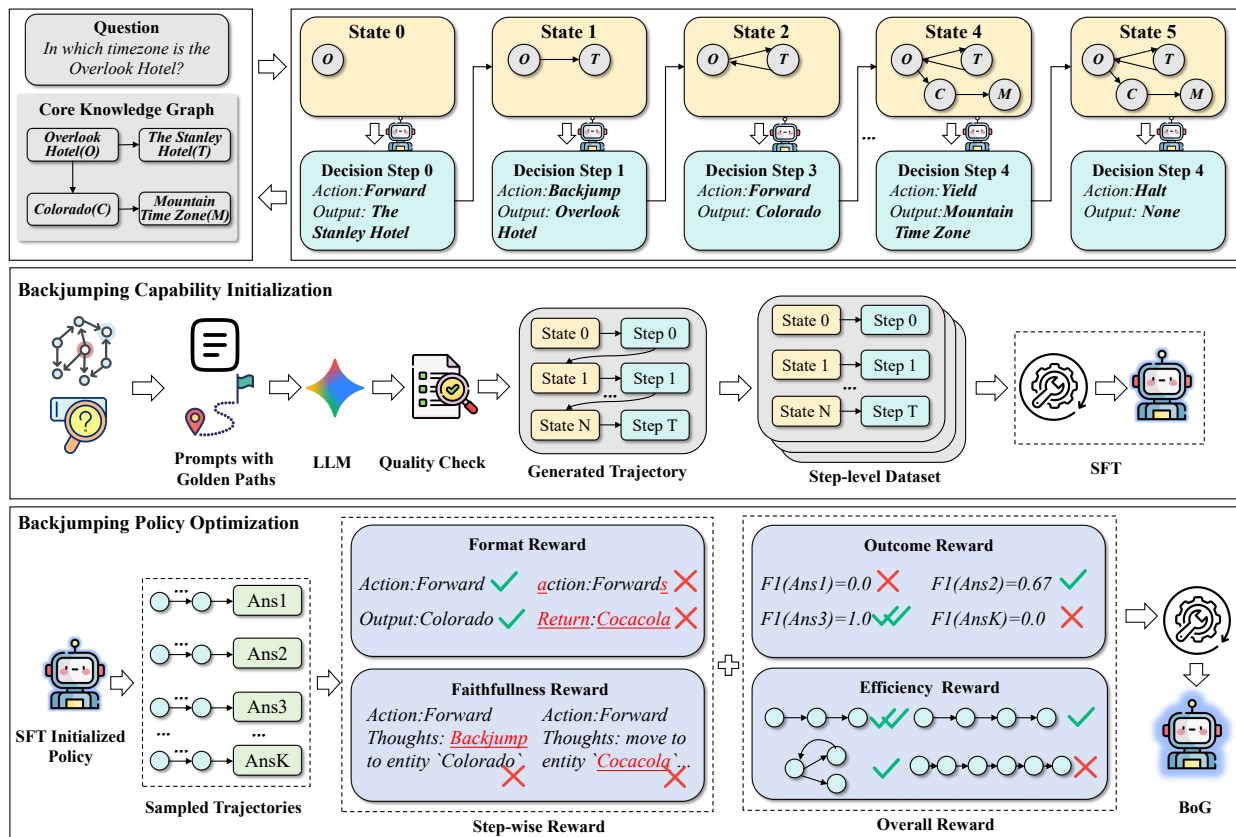

*Figure 2.* The overall framework of our approach. Top: An illustrative reasoning trajectory demonstrating the four atomic actions: *Forward, Backjump, Yield, Halt*. Middle: The Supervised Fine-Tuning (SFT) stage for instilling the agent's backjumping capabilities using synthesized data. Bottom: The Reinforcement Learning (RL) stage, including the hybrid reward function.

ure 2. To address the dead-end problem caused by the mismatch between natural language queries and KG structures, we first formalize the reasoning process into a structural scaffold using four atomic operations (Section 3.1), then instill basic backjumping capabilities through supervised fine-tuning on synthetic data (Section 3.2), and finally optimize the backjumping policy with a hybrid reward function during reinforcement learning (Section 3.3).

### 3.1. Formalization of the Reasoning Process

Recent agent-based methods treat LLMs as autonomous agents that iteratively traverse the KG by selecting neighboring entities or relations conditioned on the questions (Jiang et al., 2025; Luo et al., 2025a). By decomposing the reasoning process into sequential exploration steps, the agent can adapt its exploration strategy based on intermediate observations (Chen et al., 2024). However, existing agent-based methods often suffer from the structural mismatch between natural language questions and knowledge graph topologies. Knowledge Graphs are organized according to a strict schema, requiring every edge in a reasoning path to correspond to a predefined relation. However, user queries are

typically schema-agnostic, often describing relationships that are merely semantically similar or implicitly entailed rather than perfectly matching the canonical relations. Consequently, rigid adherence to the reasoning path implied by the question inevitably leads to reasoning impasses or erroneous conclusions due to this inherent semantic misalignment. Once dead-ends are encountered, the agent lacks an explicit mechanism to revise earlier choices, making it difficult to recover from reasoning failures.

To address this problem, we reformulate each reasoning process on the KG as an iterative cycle between the action generation phase and state transition phase. Given the environment of KG $\mathcal{G} = (\mathcal{E}, \mathcal{R})$, where $\mathcal{E}$ and $\mathcal{R}$ denote the sets of entities and relations, respectively, each reasoning step is decomposed into two distinct phases:

**Action Generation.** We denote the agent's state at the step $t$ as $s_t = \langle e_t, \mathcal{M}_t, \mathcal{N}(e_t) \rangle \in \mathcal{S}$, where $e_t$ stands for the current foothold entity, $\mathcal{M}_t$ stands for a memory unit that logs the traversed trajectory, and $\mathcal{N}(e_t) = \{(r, e_t^{Nei}) \mid (e_t, r, e_t^{Nei}) \in \mathcal{G} \vee (e_t^{Nei}, r, e_t) \in \mathcal{G}\}$ stands for $e_t$'s neighbors on the KG. Based on $s_t$ and the user's question $q$, the agent employs a policy $\pi_\theta$ to generate a decision step

$x_t = (a_t, \mathbf{R}_t, \mathcal{O}_t) \sim \pi_\theta(\cdot \mid s_t, q)$, comprising a discrete action $a_t \in \mathcal{A}$, a textual rationale $\mathbf{R}_t$ which explicates the agent's internal thinking process, and an action-specific output $\mathcal{O}_t$. To equip the agent with the ability to revert to historical nodes while preserving its fundamental reasoning capabilities, we define four atomic operations: *Forward*, *Backjump*, *Yield*, and *Halt*, thus formulate the action space as $\mathcal{A} = \{a_{\text{Forward}}, a_{\text{Backjump}}, a_{\text{Yield}}, a_{\text{Halt}}\}$. The specific definitions of these actions are as follows:

- *Forward*: Represents the standard single-hop reasoning step, where the agent selects a neighbor from $\mathcal{N}(e_t)$ as the anchor for the next step to advance the process.
- *Backjump*: Denotes a non-local reversion step. The agent abandons the current path and instantly reverts to a historical node in $\mathcal{M}_t$ as the new foothold node.
- *Yield*: Signifies the identification of a valid answer. This action allows the agent to record the current node as part of the solution set without terminating the process, thereby accommodating multi-answer queries where correct entities may appear as intermediate nodes along the path.
- *Halt*: Marks the termination of the reasoning process, signaling the agent to conclude the reasoning session.

The associated output $\mathcal{O}_t$ represents the target entity or the subset of candidate entities associated with the decision, defined according to $a_t$ as:

$$\mathcal{O}_t = \begin{cases} e_t^{Nei} & \text{if } a_t = a_{\text{Forward}} \\ e_{\text{anc}} & \text{if } a_t = a_{\text{Backjump}} \\ \mathcal{E}_t^{\text{cand}} & \text{if } a_t = a_{\text{Yield}} \\ \oslash & \text{if } a_t = a_{\text{Halt}} \end{cases} \quad (1)$$

where $e_t^{Nei}$ is the selected neighboring entity, and $e_{\text{anc}}$ denotes an ancestor entity from the preceding trajectory. $\mathcal{E}_t^{\text{cand}}$ denotes a subset of candidate entities selected from the current node and its neighbors, defined as $\mathcal{E}_t^{\text{cand}} \subseteq \{e_t\} \cup \{e' \mid (r, e') \in \mathcal{N}(e_t)\}$.

**State Transition.** Following action generation, the selected action $a_t$ triggers a deterministic transition with the knowledge graph $\mathcal{G}$. This phase is treated as a fixed post-action operation, where a transition function $\mathcal{T} : \mathcal{S} \times \mathcal{A} \rightarrow \mathcal{S}$ executes the selected action to provide objective feedback. The transition updates the agent's position and the memory $\mathcal{M}$, which together form the new state $s_{t+1}$. At each step $t$, the memory is updated as $\mathcal{M}_{t+1} = \mathcal{M}_t \cup \{(t, e_t, a_t, \mathbf{R}_t, \mathcal{O}_t)\}$. The updated memory then serves as part of the state for the next step $s_{t+1} = \mathcal{T}(s_t, a_t)$, which is defined as:

$$s_{t+1} = \begin{cases} \langle e_t^{Nei}, \mathcal{M}_{t+1}, \mathcal{N}(e_t^{Nei}) \rangle & \text{if } a_t = a_{\text{Forward}} \\ \langle e_{\text{anc}}, \mathcal{M}_{t+1}, \mathcal{N}(e_{\text{anc}}) \rangle & \text{if } a_t = a_{\text{Backjump}} \\ \langle e_t, \mathcal{M}_{t+1}, \mathcal{N}(e_t) \rangle & \text{if } a_t = a_{\text{Yield}} \\ \langle \varnothing, \mathcal{M}_{t+1}, \varnothing \rangle & \text{if } a_t = a_{\text{Halt}} \end{cases} \quad (2)$$

Upon obtaining $s_{t+1}$, the agent proceeds to the next iteration by generating a new action. This iterative exploration continues until the agent executes the halt action. Specifically, when the process terminates, the final answer set $\mathcal{E}^{\text{ans}}$ is formed by aggregating all candidate entities recorded in $\mathcal{M}_T$ through all *Yield* actions.

### 3.2. Backjumping Capability Initialization

Although the atomic operations define the structural scaffold for reasoning, a vanilla LLM lacks the inherent capability to execute these transitions effectively within a complex KG environment. To bridge this gap, we construct a synthetic dataset of structured reasoning trajectories and perform Supervised Fine-Tuning (SFT) to instill basic navigation and backjumping abilities into the agent in this section.

An intuitive approach to generate training data is utilizing a powerful LLM to explore the graph step by step through the proposed action generation and state transition phases, as described in Section 3.1. In this naive paradigm, each state transition $s_t \xrightarrow{a_t} s_{t+1}$ can be recorded and concatenated to construct a dataset of reasoning trajectories. However, without external guidance, we observe that the resulting trajectories tend to be noisy and suboptimal. Such data lacks the distinct structural features required to effectively characterize backjumping, making it impossible to instill robust backjumping capabilities into the agent.

**Oracle-Conditioned Trajectory Construction.** Therefore, we adopt an oracle-conditioned data construction strategy that leverages relevant relational paths to augment the naive step-by-step trajectory generation process. For each instance, we first obtain reference relational paths between the start entities and the answer entities using a breadth-first search (BFS) over the knowledge graph $\mathcal{G}$. We incorporate oracle-provided reference paths into the input prompt of Gemini 2.5 Flash (Comanici et al., 2025), which serves as our actor agent. Crucially, the prompt explicitly instructs the actor to use the reference paths only as structural hints during action generation, without directly relying on or revealing the ground-truth answers. This encourages the agent to perform its own reasoning at each step, while still being subtly guided by the structural hints from the reference paths. The detailed prompt templates and algorithm used to condition the actor model are provided in Appendix F.

**Supervised Fine-Tuning Objective.** We define a transition at step $t$ of trajectory $i$ as $y_t^{(i)} = (s_t^{(i)}, x_t^{(i)})$. The overall trajectory is then represented as $\tau^{(i)} = (y_0^{(i)}, y_1^{(i)}, \ldots, y_{T_i}^{(i)})$ and the entire dataset is expressed as $\mathcal{D} = \{\tau^{(i)}\}_{i=1}^N$. The SFT objective is defined as a reconstruction loss over all

transitions in all trajectories:

$$\mathcal{L}_{\text{SFT}}(\theta) = - \sum_{\tau^{(i)} \in \mathcal{D}} \sum_{t=0}^{T_i} \log \pi_\theta^{\text{SFT}}(x_t^{(i)} \mid s_t^{(i)}, q^{(i)}) \quad (3)$$

where $\pi_\theta^{\text{SFT}}$ denotes the policy after fine-tuning. Through this SFT stage, the agent acquires preliminary navigation and backjumping capabilities by imitating the reasoning trajectories provided in the dataset.

### 3.3. Backjumping Policy Optimization

Although SFT equips the agent with basic backjumping capabilities, the learned policy $\pi^{\text{SFT}}$ remains dependent on specific patterns, limiting its ability to adaptively explore in complex or unseen graph environments. To internalize these capabilities and encourage adaptive exploration, we further optimize the policy using Group Relative Policy Optimization (GRPO) (Shao et al., 2024) with a hybrid reward formulation. Specifically, the hybrid reward consists of a step-wise reward that evaluates the quality of the individual transition at each step, and an overall reward that evaluates the effectiveness of the entire exploration trajectory.

**Step-wise Reward Modeling.** To ensure the format correctness and logical validity of the exploration steps, we introduce a fine-grained reward that integrates a format reward $R_{\text{fmt}}$ and a reasoning faithfulness reward $R_{\text{faith}}$:

- *Format Reward.* Following the schema constraints, this reward encourages the agent to adhere to the intended reasoning structure. At each step $t$, we check whether the generated transition $x_t^{(i)}$ is ill-formed. Formally, let $\mathcal{F}$ be the set of all ill-formed transitions that violate the predefined structural constraints and $\mathbb{I}(\cdot)$ be the indicator function. Then, the trajectory-level format reward is defined as: $R_{\text{fmt}}(\tau) = \max\left(0, 1.0 - 0.5 \cdot \sum_{t=0}^{T} \mathbb{I}(x_t^{(i)} \in \mathcal{F})\right)$. This ensures that the agent maximizes structural correctness to retain the full format score.

- *Reasoning Faithfulness Reward.* To evaluate whether the generated reasoning trace is faithful to the current exploration context, we introduce an auxiliary verifier model that assesses the hallucination risk of $\mathbf{R}_t$ conditioned on the current state $s_t$. Concretely, at each step $t$, the verifier $V_\phi$ takes $(s_t, \mathbf{R}_t)$ as input and outputs a faithfulness score indicating whether the reasoning trace is semantically grounded in the accessible information at that state. The overall reasoning faithfulness reward for a trajectory $\tau$ is calculated as: $R_{\text{faith}}(\tau) = \sum_{t=0}^{T} V_\phi(s_t, \mathbf{R}_t)$.

**Overall Reward Modeling.** Beyond step-wise validity, the holistic quality of the exploration is evaluated based on answer correctness and reasoning efficiency. The overall reward is composed of an outcome reward $R_{\text{outcome}}$ and an efficiency reward $R_{\text{len}}$:

- *Outcome Reward.* To explicitly align policy optimization with the task objective, we introduce an outcome-based reward that evaluates the final answer's quality of an entire exploration trajectory. We parse $\mathcal{E}^{\text{ans}}$ into an entity-level prediction set and compute its $F_1$ score against the ground-truth answer set $\mathcal{E}^{\text{gold}}$. The outcome reward is computed as $R_{\text{outcome}}(\tau) = F_1(\mathcal{E}^{\text{ans}}, \mathcal{E}^{\text{gold}})$.

- *Efficiency Reward.* To prevent the agent from engaging in aimless loops or excessively long reasoning paths, we introduce a length-based efficiency reward that penalizes overly long exploration trajectories. The efficiency reward is defined as a linear decay function with respect to the number of steps taken in an episode. Let $T_{\text{len}} \geq 1$ denote the total number of steps. The efficiency reward is given by: $R_{\text{len}}(\tau) = \max(0, 1.0 - 0.1 \cdot T_{\text{len}})$.

Finally, the total reward for a trajectory $\tau$ used in GRPO is the weighted sum of these components: $R_{\text{hybrid}}(\tau) = R_{\text{outcome}}(\tau) + \alpha_1 R_{\text{len}}(\tau) + \alpha_2 R_{\text{fmt}}(\tau) + \alpha_3 R_{\text{faith}}(\tau)$.

**Reinforcement Learning Objective.** We optimize the exploration policy using GRPO. We initialize the policy $\pi_\theta^{\text{RL}}$ from the SFT-pretrained policy $\pi_\theta^{\text{SFT}}$. For a given $(\mathcal{G}, q)$, GRPO samples a group of $K$ trajectories $\{\tau_i\}_{i=1}^{K}$ from the current policy $\pi_{\theta_{\text{old}}}^{\text{RL}}$. The objective function is formulated as:

$$
\begin{aligned}
\mathcal{J}_{\text{GRPO}}(\theta) = \mathbb{E}_{\{\tau_i\}_{i=1}^{K} \sim \pi_{\theta_{\text{old}}}^{\text{RL}}} \frac{1}{K} \sum_{i=1}^{K} \frac{1}{|\tau_i|} \sum_{x_t \in \tau_i} \Big( \\
\min \left[ \psi_\theta(x_t) \hat{R}(\tau_i), \text{clip}(\psi_\theta(x_t), 1 \pm \epsilon) \hat{R}(\tau_i) \right] \\
- \beta \, \mathbb{D}_{\text{KL}}(\pi_\theta^{\text{RL}} \,||\, \pi_\theta^{\text{SFT}}) \Big)
\end{aligned}
\quad (4)
$$

Here, $\psi_\theta$ denotes the probability ratio between the current policy and the old policy for the sampled action at state $x_t$. $\hat{R}(\tau_i)$ denotes the group-relative normalized return of trajectory $\tau_i$, computed by standardizing the hybrid reward $R_{\text{hybrid}}(\tau)$ within each sampled group. Detailed computation procedures and parameter definitions of GRPO are provided in Appendix G.

## 4. Experiments

### 4.1. Experimental Setup

**Datasets and Evaluation Metrics.** To demonstrate the effectiveness of BoG on complex reasoning over knowledge graphs, we adopt three widely-used multi-hop KGQA datasets: WebQSP (Yih et al., 2016), CWQ (Talmor & Berant, 2018) and GrailQA (Gu et al., 2021). For GrailQA,

*Table 1.* Comparison of the performance of BoG against state-of-the-art methods across three KGQA benchmarks. The best results are bolded and the second-best scores are underlined. The performance of the closed-source baselines under the same agent framework is marked with [†]. Reproduced results are marked with [*]. Note that BoG$_{SFT}$ refers to the agent trained exclusively via SFT.

| Types | Methods | Model | WebQSP | | CWQ | | GrailQA | |
|---|---|---|---|---|---|---|---|---|
| | | | Hits@1 | $F_1$ | Hits@1 | $F_1$ | Hits@1 | $F_1$ |
| Closed-sourced | [†]Gemini-2.5 Flash | - | 79.2 | 51.9 | 58.4 | 52.6 | 56.5 | 62.9 |
| | [†]Gemini-2.5 Pro | - | 71.5 | 50.1 | 74.8 | 64.8 | 60.4 | 68.5 |
| | [†]GPT-5.2 | - | 66.7 | 45.4 | 68.8 | 59.5 | 64.6 | 57.9 |
| Retrieval-Based | RoG (2024) | Llama-2-7B | 85.7 | 70.8 | 62.6 | 52.6 | - | - |
| | EWEK-QA (2024) | WebGLM-10B | 71.3 | - | 52.5 | - | 60.4 | - |
| | EffiQA (2025) | GPT-4 | 82.9 | - | 69.5 | - | 78.4 | - |
| | GNN-RAG (2025) | Llama-2-7B | 85.7 | 71.3 | 66.8 | 59.4 | - | - |
| | DoG (2025) | Llama-3.1-8B | 91.4 | 77.5[*] | 76.2 | 61.1[*] | 83.4[*] | 80.7[*] |
| | GCR (2025b) | GPT-4o-mini | 92.2 | 74.1 | 72.7 | 75.8 | 84.9[*] | 79.2[*] |
| Agent-Based | ToG (2023) | ChatGPT | 76.2 | - | 57.6 | - | 68.7 | - |
| | | GPT-4 | 82.6 | - | 68.5 | - | 81.4 | - |
| | PoG (2024) | GPT-3.5 | 82.0 | - | 63.2 | - | 76.5 | - |
| | | GPT-4 | 87.3 | - | 75.0 | - | 84.7 | - |
| | KG-Agent (2025) | Llama-2-7B | 83.3 | 81.0 | 72.2 | 69.8 | 86.1 | - |
| | KBQA-o1 (2025a) | Llama-3.1-8B | 63.5[*] | 59.8 | 43.2[*] | 41.7[*] | 71.9 | 78.5 |
| | | Qwen2.5-7B | - | 57.8 | - | - | 70.8 | 77.9 |
| Agent-Based | BoG$_{SFT}$ | Qwen2.5-7B | 85.5 | 74.6 | 75.3 | 68.4 | 80.1 | 77.3 |
| | | Llama-3.1-8B | 88.5 | 77.6 | 79.4 | 71.1 | 84.6 | 78.7 |
| | BoG | Qwen2.5-7B | 91.8 | 78.9 | 81.4 | 72.0 | 87.5 | 81.2 |
| | | Llama-3.1-8B | **92.7** | **81.3** | **84.0** | **78.3** | **90.9** | **82.4** |

we adopt the same test setting as ToG (Sun et al., 2023) for evaluation. A full description of the datasets can be found in Appendix B. Following previous work (Wang et al., 2023b), we use Hits@1 and $F_1$ as evaluation metrics to assess the performance of our method.

**Baselines.** We compare our method with recent baselines, which can be broadly categorized into retrieval-based approaches (Wu et al., 2024; Luo et al., 2025a) and agent-based approaches (Wang et al., 2023b; Jiang et al., 2025). Furthermore, we include the results of closed-source LLMs, including the Gemini-2.5 Family (Comanici et al., 2025) and GPT-5 (Singh et al., 2025), as additional benchmarks. These models are evaluated under the same reasoning formalization, where our fine-tuned model is replaced by their respective APIs. The implementation details, including the API-level evaluation settings for closed-source LLMs, are provided in Appendix E.

**Implementation Details.** We perform evaluations on two open-source models, Qwen2.5-7B-Instruct (Qwen et al., 2025) and Llama-3.1-8B-Instruct (Dubey et al., 2024). We develop a pipeline that prompts Gemini-2.5 Flash (Comanici et al., 2025) to synthesize high-quality step-level datasets. For the reinforcement learning stage, we utilize the verl-agent framework (Feng et al., 2025) to implement the optimization of the exploration policy. We set the weights of the efficiency reward $\alpha_1$, format reward $\alpha_2$, and faithfulness reward $\alpha_3$ to 0.2, 0.1, and 0.1, respectively. More details

on experimental configurations and costs are available in Appendix D.

*Table 2.* Ablation studies of BoG on CWQ and WebQSP datasets.

| Model | CWQ | | WebQSP | |
|---|---|---|---|---|
| | Hits@1 | $F_1$ | Hits@1 | $F_1$ |
| BoG | **84.0** | **78.3** | **92.7** | **81.3** |
| w/o Step-wise reward | 82.7 | 76.2 | 90.4 | 79.7 |
| w/o Efficiency reward | 83.1 | 77.4 | 91.9 | 80.2 |
| w/o Entire RL stage | 79.4 | 74.1 | 88.5 | 77.6 |

## 4.2. Main Results

In this section, we compare BoG with state-of-the-art baselines to demonstrate the effectiveness of retrospective exploration. From Table 1 we have several observations. First, BoG achieves the best performance across all three datasets and consistently outperforms both retrieval-based and agent-based baselines. This result demonstrates that BoG effectively overcomes the mismatch between query-derived paths and the KG's structure through retrospective exploration. Second, BoG$_{SFT}$ and BoG consistently achieve better performance compared to closed-source models, validating the necessity of SFT to instill specialized backjumping abilities that closed-source models lack. Finally, the full BoG model outperforms BoG$_{SFT}$ in all datasets, indicating that our proposed hybrid reward function effectively facilitates BoG's backjumping policy optimization during the reinforcement

*Table 3.* Evaluation results of BoG$_{SFT}$ and BoG exploration behavior and quality on CWQ test set.

| Setting | Average Step Length(↓) | Backjumping counts (↑) | Faithfulness(↑) | Comprehensiveness(↑) |
|---|---|---|---|---|
| BoG$_{SFT}$ | 5.3 | 227 | 0.78 | 0.74 |
| BoG | 3.6 | 302 | 0.89 | 0.82 |

*Table 4.* Comparison of performance on different logical patterns and reasoning depths.

| Method | | Conjunction | Comparative | Superlative | Composition | 1-hop | 2-hop | 3-hop | ≥4-hop | Overall |
|---|---|---|---|---|---|---|---|---|---|---|
| KGQA-o1 | Hits@1 | 51.3 | 24.1 | 40.9 | 40.6 | 53.8 | 45.6 | 34.7 | 26.8 | 43.2 |
| | $F_1$ | 46.2 | 14.4 | 16.9 | 33.7 | 51.2 | 43.4 | 30.6 | 24.2 | 41.7 |
| GCR | Hits@1 | 73.1 | 66.3 | 64.4 | 66.8 | 79.0 | 71.9 | 62.1 | 47.7 | 68.2 |
| | $F_1$ | 63.7 | 57.7 | 52.6 | 59.7 | 66.3 | 63.0 | 56.6 | 45.8 | 60.3 |
| BoG | Hits@1 | **93.8** | **74.3** | **91.8** | **74.5** | **90.8** | **80.3** | **75.9** | **55.7** | **84.0** |
| | $F_1$ | **92.6** | **72.5** | **85.3** | **63.9** | **89.1** | **70.1** | **67.4** | **48.5** | **78.3** |

learning stage.

### 4.3. Ablation Study

**Ablation study of hybrid reward.** We conduct an ablation study to evaluate the core components of BoG on the CWQ and WebQSP datasets (Table 2). Removing the step-wise reward results in noticeable declines in both Hits@1 and $F_1$ scores, indicating that optimizing each individual step plays a critical role in cumulatively improving the overall reasoning performance. Furthermore, removing the efficiency reward also leads to performance degradation. We argue that the efficiency reward helps reduce repetitive backtracking and redundant reasoning, leading to more efficient exploration. Finally, BoG performs worst without the entire RL stage, confirming that while SFT provides basic navigation skills, the RL phase is the key to turning individual actions into an optimized exploration strategy.

**Analysis of exploration behavior and quality.** To analyze how the RL stage reshapes the agent's exploration dynamics, we evaluate several critical performance dimensions. First, to gauge the navigational efficiency of the agent, we compute the average step length across reasoning episodes, which serves as a proxy for the agent's ability to reach solutions with minimal redundant transitions. Second, we quantify the backjumping capability using the frequency of backjump actions executed during inference. Finally, beyond structural metrics, we assess the semantic quality of the generated reasoning trails through two critical dimensions: faithfulness (the strict adherence to valid graph facts) and comprehensiveness (the completeness of the derived answer). To ensure a rigorous and objective evaluation of these nuanced metrics, we employ the advanced closed-source model GPT-4o (Hurst et al., 2024) as an external judge, utilizing a set of specialized evaluation prompts which are detailed in Appendix H.

As shown in Table 3, BoG achieves superior path efficiency

with significantly reduced average step length. Specifically, the increased backjump frequency suggests that the hybrid reward incentivizes the agent to proactively prune unpromising branches rather than persisting in barren subgraphs. Consequently, the agent concentrates on semantically relevant structures, learning the optimal timing and destinations for state reversion. Furthermore, substantial gains in faithfulness and comprehensiveness indicate that BoG refines its policy towards rigorous logic during RL, fostering a positive feedback loop that optimizes the global exploration strategy.

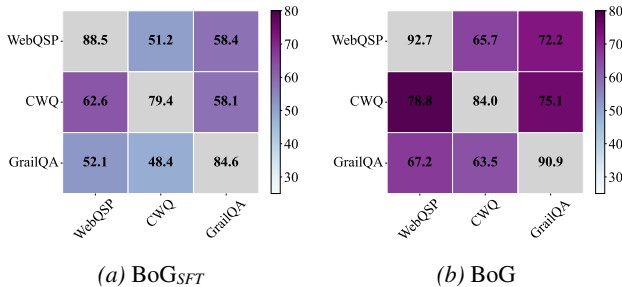

*(a)* BoG$_{SFT}$        *(b)* BoG

*Figure 3.* Comparison of Hits@1 performance across three datasets under out-of-distribution (O.O.D.) settings.

### 4.4. Performance on Different Reasoning Patterns

To further analyze the generalization of our framework, we divide the CWQ test set by logical pattern and hop count, following (Luo et al., 2025b). As shown in Table 4, BoG achieves the best performance in all logical patterns and reasoning depths. We attribute this success to our history-aware backjumping mechanism, which transforms the reasoning process from a fragile, one-way path into a flexible trial-and-error exploration. In complex scenarios involving intricate relationship combinations or deep paths (≥ 3 hops), BoG's backjumping allows the agent to revisit previous decision points when it identifies a dead end or a logical mismatch.

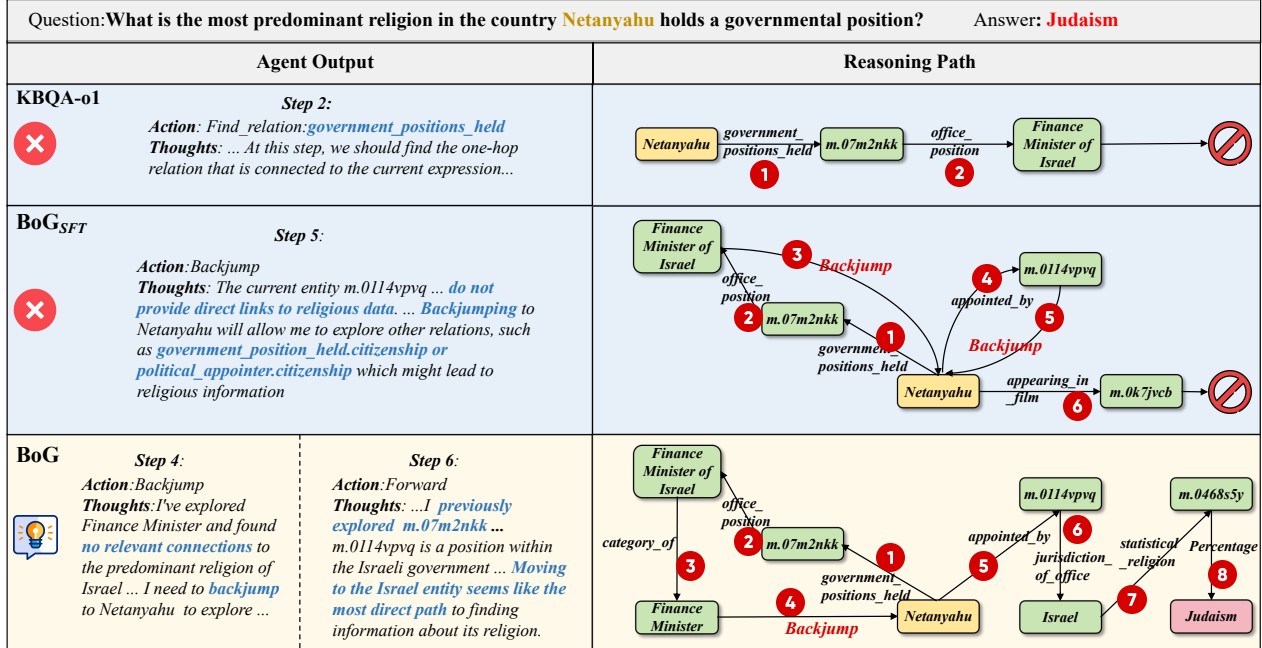

*Figure 4.* A case study comparing KBQA-o1, BoG$_{SFT}$ and BoG on a real example from the CWQ dataset. The left panels present the agent's output during critical decision-making, with core reasoning logic highlighted in bold blue. The right panels visualize the corresponding exploration paths, where numbered circles indicate the chronological sequence of the reasoning steps. For better visualization, relations are undirected and simplified.

This capability enables the agent to actively explore alternative paths and gather more context from the KG. By effectively self-correcting its trajectory, BoG can successfully navigate through exponentially expanding search spaces to find the correct answer, leading to a significant performance margin over fixed-path models on complex reasoning scenarios.

### 4.5. Performance on Out-of-Domain Dataset

As shown in Figure 3, the heatmaps illustrate the model's performance across various training-testing combinations, where the vertical axis ($Y$-axis) represents the training dataset and the horizontal axis ($X$-axis) denotes the evaluation (test) dataset. Notably, BoG consistently outperforms BoG$_{SFT}$ across three O.O.D. datasets, demonstrating that the hybrid reward enables BoG to learn transferable search logic—particularly in terms of backjumping timing and depth, which generalizes effectively to novel graph environments. This allows the agent to maintain performance stability by prioritizing strategic navigation over the imitation of fixed training trajectories.

### 4.6. Case Study

Figure 4 illustrates a real example from the CWQ test set. During exploration, KBQA-o1 is misled by a highly question-aligned but deceptive relation "*government_positions_held*". This relation appears seman-

tically relevant to the query but directs the agent toward a wrong reasoning path, therefore leading the reasoning process to a failure. BoG$_{SFT}$ demonstrates a limited ability to backjump when encountering dead-end entities. However, its reasoning process remains constrained by previously imitated exploration trajectories. Consequently, at step 5, BoG$_{SFT}$ prematurely terminates further exploration due to the absence of familiar or supervised relation patterns, preventing it from discovering alternative reasoning paths that could lead to the correct answer.

Finally, our BoG model exhibits a more adaptive and robust exploration strategy. The agent first recognizes that the current branch is unproductive and actively backjumps to the initial entity. Leveraging accumulated error experience, it then expands exploration toward previously underexplored relation patterns, ultimately redirecting the reasoning process to the "*Israel*" entity and its associated religious statistics. From this example, BoG demonstrates a clear advantage by combining enhanced retrospective exploration with historical error accumulation, allowing the agent to explore the graph more aggressively, recover from failed explorations through timely backjumping, and utilize memory to progressively converge to the correct answer.

## 5. Conclusion

In this paper, we propose a novel framework named Backjump-on-Graph to address topological dead-ends in

agentic KG navigation. By formalizing reasoning into atomic operations and leveraging reinforcement learning, BoG empowers LLMs to perform history-aware backjumping and effectively explore alternative reasoning paths. Extensive experimental results on three KBQA benchmarks demonstrate that BoG significantly outperforms existing baselines, validating the effectiveness of our methods in complex reasoning tasks.

## Acknowledgement

This work was supported by National Cyber Security-National Science and Technology Major Project (2025ZD1502200) and the Henan Provincial Major Industrial Key Technology Research Open Bidding for Leading Talents Project (251000210400).

## Impact Statement

This work proposes Backjump-on-Graph (BoG), an agentic framework for Knowledge Graph Question Answering (KGQA) that enables Large Language Models (LLMs) to recover from exploration failures through retrospective navigation. By formalizing reasoning into four atomic operations and optimizing the policy via GRPO, BoG effectively mitigates navigation traps where user queries do not directly align with the underlying graph schema. While BoG improves the accuracy of multi-hop reasoning, its performance remains dependent on the initial coverage of the knowledge graph and the quality of reasoning traces. To address these challenges, we propose four future research directions: (1) extending BoG to handle large-scale, heterogeneous graphs with evolving schemas, (2) improving the model's generalization across low-resource domains such as cybersecurity and clinical medicine, (3) integrating multi-modal knowledge sources to support more complex reasoning tasks, and (4) enhancing the interpretability of the backjumping mechanism for better error diagnosis in mission-critical applications. This research does not present foreseeable ethical risks as it relies on public benchmarks. By improving the transparency of graph-based reasoning, this work fosters the development of more trustworthy AI systems.

## Limitations

While BoG provides a novel RL-based paradigm for retrospective KG exploration, it has a few limitations that point to future directions:

**Dependency on Oracle Entity Linking.** Following standard KGQA protocols, BoG assumes correct initial entity linking. In real-world scenarios with noisy queries, linking errors will propagate to the reasoning phase. Future work will explore end-to-end frameworks that jointly handle disambiguation and reasoning.

**Static Reward Weighting.** The coefficients in our hybrid reward function are empirically fixed. While they generalize well across our evaluated benchmarks, they may lack optimal adaptability for entirely different KG schemas. Developing an adaptive, learnable reward weighting mechanism is a promising direction.

**Context Limits in Extreme Horizons.** While our trajectory memory ($M_t$) is highly efficient for standard reasoning depths, extremely complex queries requiring dozens of backjumps could eventually saturate the context windows of 7B/8B models. Future iterations will explore dynamic memory pruning to scale to ultra-long-horizon reasoning.

**Training Computational Overhead.** Using an LLM-based verifier for step-wise faithfulness rewards during GRPO rollouts introduces additional computational cost compared to standard SFT. However, since this verifier is entirely decoupled from the inference phase, the model's deployment efficiency remains completely unaffected.

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

# A. Preliminary

**Knowledge Graphs (KGs).** A knowledge graph (KG) is a directed multi-relational graph composed of factual triples. We denote a KG as $\mathcal{G} = (\mathcal{E}, \mathcal{R}, \mathcal{U})$, where $\mathcal{E}$ and $\mathcal{R}$ are the sets of entities and relations, and $\mathcal{U} \subseteq \mathcal{E} \times \mathcal{R} \times \mathcal{E}$ is the set of relational triples. Each triple $(s, r, o) \in \mathcal{U}$ represents a factual statement that connects a subject entity $s$ to an object entity $o$ via relation $r$.

**Reasoning Paths.** A reasoning path is an ordered sequence of connected triples in $\mathcal{G}$ that forms an interpretable chain from a start entity to a target entity:

$$P = e_0 \xrightarrow{r_1} e_1 \xrightarrow{r_2} \cdots \xrightarrow{r_L} e_L, \tag{5}$$

where $L$ is the path length and each step $(e_{i-1}, r_i, e_i) \in N$ is a valid triple in the KG.

**Problem Formulation.** Given a natural language question $q$ and a KG $\mathcal{G}$, the goal of KGQA is to predict the correct answer set $A$:

$$\text{Ans} = f(q, \mathcal{G}). \tag{6}$$

Following prior work (Sun et al., 2019; Luo et al., 2024), we assume that the entities mentioned in $q$ and the answer entities are linked to their corresponding KG entities in $\mathcal{E}$.

# B. Datasets

**WebQSP.** WebQSP is a widely-used KGQA benchmark that focuses on answering simple, fact-based questions, which usually require retrieving a single fact from the knowledge graph. Each question is accompanied by a complete SPARQL query that can be directly executed on Freebase to obtain the answers.

**CWQ.** CWQ presents higher difficulty compared to WebQSP by introducing questions that require compositional reasoning. These questions typically involve multiple constraints, conjunctions, or superlative operations, and thus require multi-hop or multi-relation reasoning over the Freebase knowledge graph. The dataset provides complex SPARQL annotations that explicitly capture these compositional reasoning patterns.

**GrailQA.** GrailQA is a large-scale and high-quality KGQA dataset constructed on Freebase, aiming to evaluate the generalization ability of models across three distinct levels: i.i.d., compositional, and zero-shot. This design enables a fine-grained analysis of a model's capability to handle seen patterns, novel combinations of known patterns, and entirely unseen domains or relations.

# C. Baselines

We compare our proposed model with EoG and a range of representative KG-enhanced LLM-based reasoning methods.

**EWEK-QA.** EWEK-QA (Dehghan et al., 2024) focuses on efficiently integrating two types of external knowledge: web text and knowledge graphs. It consists of an adaptive web retriever for dynamically extracting text snippets and a knowledge graph retriever (TOG-E) for structured information. This design improves answer accuracy across multiple QA benchmarks while being faster.

**ToG (Think-on-Graph).** ToG (Sun et al., 2023) formulates KGQA as an agent-based reasoning process, where the LLM actively interacts with the knowledge graph. Instead of directly translating questions into queries, the model performs beam search over the KG to iteratively explore relevant entities and relations. This paradigm enhances the reasoning ability of smaller models on complex knowledge-based questions.

**EffiQA.** EffiQA (Dong et al., 2025) addresses the trade-off between performance and efficiency in KG-enhanced LLMs. It proposes an "LLM-as-planner, small-model-as-executor" framework, where a lightweight plugin model handles graph traversal and semantic pruning. This design reduces inference cost while maintaining competitive accuracy on KBQA benchmarks.

**RoG (Reasoning on Graphs).** RoG (Luo et al., 2024) is based on the observation that knowledge graph relations form faithful reasoning paths. Through fine-tuning, the LLM is guided to generate relation-level reasoning plans that can be verified against the KG. These plans are then grounded by retrieving concrete path instances from the graph, which are used for final answer inference, resulting in interpretable and faithful reasoning.

**GNN-RAG.** GNN-RAG (Mavromatis & Karypis, 2025) combines GNN-based structural reasoning with LLM-based language understanding for KBQA. A GNN operates over a subgraph to identify answer candidates and corresponding reasoning paths, which are subsequently provided to the LLM as contextual evidence. This hybrid approach demonstrates strong performance on benchmarks such as WebQSP and CWQ.

**DoG (Decoding on Graphs).** DoG (Li et al., 2025) introduces the concept of a *well-formed chain* to constrain LLM generation to valid and connected knowledge graph triples. It constructs a Trie from the local KG to dynamically mask invalid tokens during decoding, ensuring that all generated reasoning steps remain grounded in the graph.

**KG-Agent.** KG-Agent (Jiang et al., 2025) adopts a grey-box agent framework by fine-tuning a smaller LLM as an expert tool user for KG operations. The agent learns to generate executable programs composed of predefined tool calls for KG access and logical operations, trained on a large-scale dataset synthesized from existing logical forms.

**GCR (Graph-Constrained Reasoning).** GCR (Luo et al., 2025b) enforces graph-constrained reasoning by guiding LLM decoding with KG structures. It encodes KG reasoning paths into a KG-Trie index and employs a lightweight KG-specialized LLM during inference. This design ensures faithful reasoning and allows the model to generalize to unseen knowledge graphs without additional training.

**KGQA-o1.** KGQA-o1 (Luo et al., 2025a) introduces an agentic framework that integrates Monte Carlo Tree Search (MCTS) to optimize logical form generation. By employing a ReAct-based agent equipped with atomic query tools, the model heuristically explores the KB environment under the guidance of policy and reward models. Additionally, it utilizes incremental fine-tuning with self-generated data to reduce reliance on human annotation while enhancing performance in low-resource settings.

**PoG.** PoG (Chen et al., 2024) introduces a self-correcting adaptive planning paradigm to address the limitations of fixed-breadth exploration in KG-augmented LLMs. It operates by decomposing questions into sub-objectives and employing three synergistic mechanisms—Guidance, Memory, and Reflection—to orchestrate the reasoning process. This design allows the model to dynamically adjust exploration breadth and backtrack to correct erroneous reasoning paths, ensuring faithful and efficient graph reasoning.

## D. Detailed Experimental Setup

To evaluate the generality of our approach, we conduct experiments on Llama-3.1-8B-Instruct (Dubey et al., 2024) and Qwen2.5-7B-Instruct (Qwen et al., 2025). In the supervised fine-tuning stage, we use Gemini-2.5-Flash to synthesize action-based data for fine-tuning. In the reinforcement learning stage, we adopt Group Relative Policy Optimization (GRPO) implemented in Verl-agent (Feng et al., 2025) framework.

For supervised fine-tuning, we train for 3 epochs on a single node with $8\times$ H100 GPUs, using a per_device_train_batch_size of 2 and a learning rate of 1e-5 with a cosine annealing schedule. Checkpoints are saved every 100 steps. For instance, we train for a total of 960 steps and select the best-performing checkpoint on the validation set for subsequent stages.

For GRPO training, we set the policy learning rate to $4 \times 10^{-6}$ and sample six responses per prompt, following the default GRPO configuration in Verl-agent (Feng et al., 2025). Rollouts are performed using vLLM with a tensor parallel size of one and a GPU memory utilization ratio of 0.6, with a sampling temperature of 0.8 and a top-$p$ value of 1.0. During trajectory generation, rollouts are truncated after at most 15 steps for training and 20 steps for evaluation. Training is conducted on a single node with $8\times$ H100 GPUs, using a mini-batch size of 512 and a micro-batch size of 8. The maximum sequence length is set to 10,000 tokens, with a maximum response length of 1,024 tokens. To reduce GPU memory consumption, we enable gradient checkpointing and adopt FSDP with CPU offloading. Model checkpoints are saved every 20 training steps. If training becomes unstable, evaluation is performed using the most recent stable checkpoint identified by the training reward trajectory; otherwise, we report results from the final checkpoint.

Rewards are computed using a hybrid reward function consisting of the following components:

- **Format Reward** ($R_{\mathbf{fmt}}$)**:** The agent output is parsed according to the predefined key–value structure. A format error is triggered if an undefined key or an invalid value type is detected.

- **Reasoning Faithfulness Reward** ($R_{\mathbf{faith}}$)**:** The input context and the generated output are concatenated and fed into a frozen Llama-3.1-8B model to produce a score that measures the degree of hallucination.

- **Outcome Reward** ($R_{\text{outcome}}$)**:** When the agent executes a halt action or reaches the maximum step limit, the predicted answer is extracted from the content enclosed by the `answer` tag. The reward is computed via entity-level matching against the ground-truth answer.

- **Efficiency Reward** ($R_{\text{len}}$)**:** This reward is determined solely by the total trajectory length, i.e., the number of steps taken within an episode.

**Hyperparameters and Configuration for GRPO Phases.** This section outlines the hyperparameter selection and system configurations employed during the RL stage using Group Relative Policy Optimization. All RL experiments were conducted within the Verl-agent framework (Feng et al., 2025), utilizing a single compute node equipped with eight NVIDIA H100 GPUs. The policy network was initialized from the post-SFT checkpoint. To optimize hardware efficiency, we employed gradient checkpointing and Fully Sharded Data Parallel (FSDP). Specifically, both model parameters and optimizer states were offloaded to the CPU to minimize memory overhead and maximize training throughput.

Regarding data processing, the model was trained with a global batch size of 512. We constrained the maximum sequence lengths to 10,000 tokens for prompts and 1,024 tokens for generated responses. The actor network was optimized using a learning rate of $4 \times 10^{-6}$. Each iteration of rollout collection was followed by two internal PPO optimization epochs. Notably, KL regularization was omitted by disabling the KL loss term unless otherwise specified.

Exploration was facilitated through a dedicated rollout procedure integrated with the vLLM engine. For each input prompt, we generated six candidate responses via temperature sampling at a temperature of 0.8 and a top-p of 1.0 to ensure trajectory diversity. To maintain response conciseness, we incorporated a penalty for excessive length through an overlength buffer. The entire RL process spanned five training epochs. Furthermore, under the optimal configuration for the CWQ dataset, the model demonstrated rapid convergence within 96 optimization steps, totaling 562.4 GPU-hours.

## E. Closed-source LLM Evaluation Reproducibility Details

This section details the evaluation protocol for assessing KBQA datasets with commercial closed-source large language models (LLMs), with an emphasis on reproducibility and fair comparison. We will introduce precise specification of the prompt engineering strategies, hyperparameters, and the interaction environment. We set the sampling temperature to 0.2 and enforce a timeout of 120 seconds to accommodate longer generations induced by explicit reasoning. The Knowledge Graph (KG) access was mediated via a SPARQL endpoint querying a Freebase dump. For semantic filtering, we utilized a local Sentence-BERT model to compute cosine similarity between the query and relation labels. Furthermore, We implement a unified evaluation pipeline in which the prompt used for commercial-model inference is identical to the one employed in the RL training phase as shown in Figure 6. Finally, The core logic of the agent combines graph traversal with LLM-based decision-making. The process is formalized in Algorithm 1. The agent maintains a state $S_t = (e_t, \mathcal{H}_t, \mathcal{N}_t)$, where $e_t$ is the current entity, $\mathcal{H}_t$ is the reasoning history (notebook), and $\mathcal{N}_t$ is the set of observed neighbors. At each step, the Semantic Filter function ($\Phi$) reduces the search space if the number of neighbors exceeds $K$.

## F. Oracle-Conditioned Trajectory Construction Details

**Oracle-Conditioned Trajectory Construction Prompt.** The template for constructing the oracle-conditioned trajectory dataset used for fine-tuning is shown in Figure 5. To facilitate the generation of high-quality reasoning paths, this prompt implements an oracle-conditioned strategy. It incorporates reference relational paths, identified via Breadth-First Search (BFS), as structural hints within the input. Crucially, the prompt instructs the model to leverage these hints to guide its exploration without directly revealing the ground-truth answers, thereby encouraging the generation of reasoning processes that are both authentic and structurally correct.

**Oracle-Conditioned Trajectory Construction Process.** The Oracle-Conditioned data construction algorithm for SFT is shown in Algorithm 2. The Oracle-Conditioned Trajectory Construction algorithm serves as a teacher-forcing data generation pipeline that synthesizes expert reasoning paths for knowledge graph navigation by utilizing ground-truth answers to guide every decision. Starting from an initial entity, the system iteratively constructs a state representing the current question, local neighborhood, and exploration memory, while a teacher policy provides the optimal action, reasoning logic, and target object to ensure the path remains aligned with the gold-standard answers. The process dynamically manages exploration by selecting promising neighbors, marking potential answers, or executing a backjump to prune dead-ends from the search space, ultimately recording each step into a structured trajectory that teaches a model both successful navigation

---

**Algorithm 1** LLM-Driven Graph Exploration and Reasoning

---

**Require:** Question $Q$, Start Entities $E_{start}$, Knowledge Graph $\mathcal{G}$, Max Steps $T_{max}$
**Ensure:** Reasoning Path $P$, Collected Answers $A_{collected}$

1: **Initialization:**
2: Initialize Path $P \leftarrow [E_{start}[0]]$
3: Initialize Notebook $\mathcal{H} \leftarrow$ "Start of search..."
4: Initialize Collected Answers $A_{collected} \leftarrow \emptyset$
5: Initialize Backjumped Set $B \leftarrow \emptyset$
6: **for** $t = 1$ **to** $T_{max}$ **do**
7:    $e_{curr} \leftarrow P.\text{last}()$
8:    *// Step 1: Perception*
9:    $neighbors \leftarrow \text{QuerySPARQL}(\mathcal{G}, e_{curr})$
10:    **if** $|neighbors| > K$ **then**
11:      $neighbors \leftarrow \Phi(neighbors, Q, K)$ {Filter by semantic similarity}
12:    **end if**
13:    Remove nodes in $B$ from $neighbors$
14:    *// Step 2: Decision Making*
15:    $prompt \leftarrow \text{ConstructPrompt}(Q, e_{curr}, neighbors, \mathcal{H}, A_{collected})$
16:    $response \leftarrow \text{LLM}(prompt)$ {Output includes Action, Entity, Reasoning}
17:    Parse JSON action $a_t$, target entity $e_{next}$, notebook update $u_t$
18:    *// Step 3: State Transition*
19:    Update Notebook: $\mathcal{H} \leftarrow \mathcal{H} + u_t$
20:    **if** $a_t ==$ 'mark_answer' **then**
21:      $A_{collected} \leftarrow A_{collected} \cup \{e_{next}\}$
22:    **else if** $a_t ==$ 'select_neighbor' **then**
23:      $P.\text{append}(e_{next})$
24:    **else if** $a_t ==$ 'backjump' **then**
25:      $B.\text{add}(e_{curr})$ {Prune current branch}
26:      $P.\text{pop}()$
27:      Ensure $P$ is not empty; otherwise terminate
28:    **else if** $a_t ==$ 'finish_search' **then**
29:      **break**
30:    **end if**
31: **end for**
32: **Return** $P, \mathcal{H}, A_{collected}$

---

and error-recovery strategies.

## G. GRPO Details

In this section, we present the details of GRPO.

### G.1. Trajectory Sampling and Notation

Given a graph-question pair $(\mathcal{G}, q)$, the exploration policy $\pi_{\theta_{\text{old}}}^{\text{RL}}$ samples a group of $K$ trajectories $\{\tau_i\}_{i=1}^K$, where each trajectory is defined as

$$\tau_i = \{x_{i,1}, a_{i,1}, \ldots, x_{i,T_i}, a_{i,T_i}\},$$

and $T_i = |\tau_i|$ denotes the trajectory length. Each state $x_{i,t}$ consists of the current query context, graph position, memory buffer, and the set of available actions. A trajectory terminates when the agent executes the *Halt* action or reaches the maximum reasoning horizon.

### G.2. Group-Relative Reward Normalization

Given a sampled group $\{\tau_i\}_{i=1}^K$, we compute the group mean and standard deviation of the hybrid rewards:

$$\mu_R = \frac{1}{K} \sum_{i=1}^K R_{\text{hybrid}}(\tau_i), \quad \sigma_R = \sqrt{\frac{1}{K} \sum_{i=1}^K \left(R_{\text{hybrid}}(\tau_i) - \mu_R\right)^2}. \tag{7}$$

The normalized group-relative return for each trajectory is defined as:

$$\hat{R}(\tau_i) = \frac{R_{\text{hybrid}}(\tau_i) - \mu_R}{\sigma_R + \delta}, \tag{8}$$

where $\delta$ is a small constant for numerical stability. This normalization removes absolute reward scale dependency and serves as a variance-reduced advantage estimator.

### G.3. Token-Level Policy Ratio

For each state–action pair $(x_t, a_t)$ along trajectory $\tau_i$, the policy ratio is computed as:

$$\psi_\theta(x_t) = \frac{\pi_\theta^{\text{RL}}(a_t \mid x_t)}{\pi_{\theta_{\text{old}}}^{\text{RL}}(a_t \mid x_t)}. \tag{9}$$

The same trajectory-level normalized return $\hat{R}(\tau_i)$ is assigned to all time steps within the trajectory.

### G.4. GRPO Objective

The GRPO optimization objective is given by:

$$\mathcal{J}_{\text{GRPO}}(\theta) = \mathbb{E}_{\{\tau_i\}_{i=1}^K \sim \pi_{\theta_{\text{old}}}^{\text{RL}}} \left[ \frac{1}{K} \sum_{i=1}^K \frac{1}{|\tau_i|} \sum_{x_t \in \tau_i} \left( \min \left[\psi_\theta(x_t)\hat{R}(\tau_i), \quad \text{clip}(\psi_\theta(x_t), 1 \pm \epsilon)\hat{R}(\tau_i)\right] \right) \right] - \beta \, \mathbb{D}_{\text{KL}}\left(\pi_\theta^{\text{RL}} \| \pi_\theta^{\text{SFT}}\right) \tag{10}$$

where $\epsilon$ controls the clipping range and $\beta$ weights the KL regularization term that constrains the learned policy to remain close to the SFT-initialized policy.

### G.5. GRPO Training implements

To facilitate the Backjumping Policy Optimization via Group Relative Policy Optimization (GRPO), we designed the prompt shown in Figure 6 and the algorithm in Algorithm 3. This template is specifically engineered to align with our hybrid reward formulation (§3.3). It imposes strict structural constraints to ensure the computability of the format reward ($R_{\text{fmt}}$) and

enforces explicit, grounded reasoning steps to enable effective hallucination assessment by the verifier for the faithfulness reward ($R_{\text{faith}}$). Furthermore, the prompt instructions encourage the agent to internalize adaptive exploration strategies, transitioning from the imitation-based behavior of SFT to the autonomous goal-seeking required to maximize the outcome ($R_{\text{outcome}}$) and efficiency ($R_{\text{len}}$) rewards.

## H. Reasoning Process Evaluation Prompts

We additionally perform LLM-as-a-judge evaluation on the model's intermediate reasoning traces and notebook memories. For each step, we provide the judge model with the original system prompt, user question, the model's action, the reasoning, and the corresponding notebook update, and ask it to produce scalar scores and free-form justifications.

### H.1. Faithfulness Evaluation

We measure whether the reasoning and notebook_update are based on accurate and verifiable facts. The prompt also requires a textual explanation specifying which facts are correct, which are inaccurate or unverifiable, and why the assigned score is appropriate, as well as whether the notebook update remains consistent with the reasoning. The full prompt is provided in Figure 8.

### H.2. Comprehensiveness Evaluation

We measure whether the thinking considers all important aspects and is thorough. Again, the judge outputs two scores in $[0, 10]$, for the reasoning and the notebook update respectively, The judge must justify its scores by indicating which important aspects are covered, which are missing, and how deep or shallow the reasoning is, and by commenting on the alignment between the notebook update and the underlying reasoning. The full prompt is provided in Figure 7.

---

**Algorithm 2** Oracle-Conditioned Trajectory Construction (Teacher Forcing)

---

**Require:** Question $q$, Knowledge Graph $\mathcal{G}$, Start Entity $e_{start}$, Ground Truth Answers $\mathcal{E}^{\text{ans}}_{gold}$, Max Steps $T_{max}$
**Ensure:** Structured Reasoning Trajectory $\tau$
  **Initialization:**
  $e_0 \leftarrow e_{start}$
  $\mathcal{M}_0 \leftarrow \varnothing$ {Initialize exploration memory (Notebook)}
  $\mathcal{V}_{pruned} \leftarrow \varnothing$ {Set of pruned dead-end nodes}
  $\tau \leftarrow \varnothing; t \leftarrow 0$
  **while** $t < T_{max}$ **do**
    *// Phase 1: State Construction & Oracle Generation*
    Retrieve neighbors $\mathcal{N}(e_t)$ from $\mathcal{G}$
    Filter valid neighbors: $\mathcal{N}(e_t) \leftarrow \mathcal{N}(e_t) \setminus \mathcal{V}_{pruned}$
    Construct state $s_t \leftarrow \langle q, e_t, \mathcal{M}_t, \mathcal{N}(e_t) \rangle$
    *// Teacher Agent generates action conditioned on Ground Truth*
    $(a_t, \mathbf{R}_t, \mathcal{O}_t) \leftarrow \pi_{\text{teacher}}(s_t \mid \mathcal{E}^{\text{ans}}_{gold})$
    *// Phase 2: State Transition $\mathcal{T}(s_t, a_t)$*
    **if** $a_t = a_{\text{mark}}$ **then**
      $\mathcal{E}^{\text{cand}}_t \leftarrow \mathcal{O}_t$ {Commit candidate entities}
      *Note: Agent remains at $e_t$*
    **else if** $a_t = a_{\text{select}}$ **then**
      $e_{t+1} \leftarrow \mathcal{O}_t$ {Navigate to neighbor $e^{Nei}_t$}
    **else if** $a_t = a_{\text{back}}$ **then**
      $\mathcal{V}_{pruned} \leftarrow \mathcal{V}_{pruned} \cup \{e_t\}$ {Permanently prune dead-end}
      $e_{t+1} \leftarrow \mathcal{O}_t$ {Backjump to ancestor $e_{\text{anc}}$}
    **else if** $a_t = a_{\text{halt}}$ **then**
      **break** {Terminate exploration}
    **end if**
    *// Update Memory and Record Step*
    $\mathcal{M}_{t+1} \leftarrow \mathcal{M}_t \cup \{(t, e_t, a_t, \mathbf{R}_t, \mathcal{O}_t)\}$
    Append $(s_t, a_t, \mathbf{R}_t, \mathcal{O}_t)$ to $\tau$
    $t \leftarrow t + 1$
  **end while**
  **Return** $\tau$

---

---

**Algorithm 3** BoG Policy Optimization via GRPO (Detailed Description for Rollout)

---

**Input:** Dataset $\mathcal{D}$, Graph $\mathcal{G}$, Policy $\pi_\theta$, Group Size $G$, Steps $T_{max}$
**for** each training iteration **do**
    Sample batch $\mathcal{B} = \{q_1, \ldots, q_B\} \sim \mathcal{D}$
    **Phase 1: Group Sampling (Rollout)**
    **for** each $q \in \mathcal{B}$ **do**
        $\mathcal{O}_q \leftarrow \varnothing$;
        **for** $k = 1$ to $G$ **do**
            $s_0 \leftarrow (e_{start}, \varnothing)$; $\tau_k \leftarrow []$
            **for** $t = 1$ to $T_{max}$ **do**
                $C_t \leftarrow$ ConstructPrompt$(q, s_{t-1}, \mathcal{G})$; $a_t \sim \pi_\theta(\cdot|C_t)$
                **if** $a_t = $ SELECT_NEIGHBOR$(e')$ **then**
                    $s_t \leftarrow$ UpdateState$(s_{t-1}, e')$
                **else if** $a_t = Backjump$ **then**
                    $s_t \leftarrow$ RevertState$(s_{t-1})$
                **else if** $a_t \in \{$FINISH, MARK_ANSWER$\}$ **then**
                    **break**
                **end if**
                Append $(C_t, a_t)$ to $\tau_k$
            **end for**
            Calculate $r_k$; $\mathcal{O}_q \leftarrow \mathcal{O}_q \cup \{(\tau_k, r_k)\}$
        **end for**
    **end for**
    **Phase 2 & 3: Advantage Estimation & Optimization**
    **for** each $q \in \mathcal{B}$ **do**
        $\mu_q, \sigma_q \leftarrow$ stats$(r_{1:G})$; $A_k \leftarrow (r_k - \mu_q)/(\sigma_q + \epsilon)$ {Group Advantage}
    **end for**
    Update $\theta$ by maximizing $\mathcal{J} = \mathbb{E}_{q \in \mathcal{B}}[\frac{1}{G} \sum_{k=1}^{G} \sum_t \min(\frac{\pi_\theta}{\pi_{old}} A_k, \text{clip}(\ldots)A_k)]$
**end for**

---

---

**SFT Data Synthesis System Prompt**

==System prompt==

You are an expert Graph Exploration Agent.
Your goal is to navigate a knowledge graph to find the correct answer(s) to the user's question.
**CRITICAL INSTRUCTION - METACOGNITIVE SEPARATION:**
- **Simulate a natural, exploratory reasoning process** as if you are discovering answers through graph navigation alone.
- **Example of GOOD reasoning:** "Relation R looks promising because it connects to entities that might be relevant to the question. Let me explore this path."
- **Example of BAD reasoning:** "I know entity X is in the target answers, so I'll mark it."
- **Example of GOOD notebook_update:** "Explored entity X via relation R. Found several connected entities that seem related to the question topic. Continuing to investigate these connections."
- **Example of BAD notebook_update:** "Found entity X which is in the target answers."

**Your Mission:**
1. If the current entity OR ANY of its NEIGHBORS are Target Answers, you MUST use 'mark_answer'. **You can output a LIST of entity IDs to mark multiple answers at once.**
2. If you hit a dead end, Backtrack.
3. If you have found ALL Target Answers, use 'finish_search'.

**You are creating a "gold standard" path.** Make optimal decisions based on the available neighbors to reach the targets, but present your reasoning as natural discovery.
**Efficiency Rule:** If multiple answers are present as neighbors, mark ALL of them in a SINGLE 'mark_answer' action.

You must decide on ONE of four actions:
1. "select_neighbor": Move to a neighbor entity that brings you closer to the Target Answers.
   - For this action, 'entity' must be a list format with length 1: ["entity_id"]
   - **CRITICAL RULE:** The selected entity MUST be one of the entities listed in "Available neighbor entities" section. You CANNOT select an entity that is not in the current neighbors list.
2. "backtrack": Go back if the current branch does not lead to any missing Target Answers.
   - For this action, 'entity' must be a list format with length 1: ["entity_id"]
   - **CRITICAL CONSTRAINT:** You **CANNOT** backtrack to the current node. The `previous_node_id` must be a different node that appears earlier in your path. Backtracking to the current node will not change your position and will cause an infinite loop.
   - **CRITICAL RULE:** Once you backtrack FROM a node, that node is PERMANENTLY EXCLUDED. You CANNOT select it again as a neighbor in future steps. Treat it as if it no longer exists.
3. "mark_answer": **REQUIRED** if Current Entity OR any Neighbors are in the Target Answers list.
   - **IMPORTANT: For 'mark_answer', 'entity' MUST be a LIST format: ["id1", "id2", ...]**
   - Even if marking only one entity, use list format: ["id1"]
   - You can mark multiple answers at once by including all IDs in the list.
4. "finish_search": **REQUIRED** if you have found ALL Target Answers in the list.
   - **IMPORTANT CONSTRAINT:** If the last step in your notebook was a 'mark_answer' action, you MUST:
     a) Ensure your 'reasoning' and 'notebook_update' do NOT repeat the same content as the previous 'mark_answer' step
     b) Provide NEW insights or confirmation that all targets are found, rather than repeating previous statements
   - For this action, 'entity' MUST be a LIST format containing ALL collected answers: ["id1", "id2", ...]
   - Include ALL entity IDs that you have marked as answers during the search process

**CRITICAL: Your response MUST be valid JSON format ONLY. Do NOT include any text before or after the JSON object.
Output ONLY the JSON object.**

Your response must be in JSON format:
{
    "action": "select_neighbor" | "backtrack" | "mark_answer" | "finish_search",
    "entity": ["entity_id"] (for select_neighbor/backtrack, must be a list with length 1) OR ["id1", "id2"] (for mark_answer) OR ["id1", "id2", ...] (for finish_search, must contain ALL collected answers),
    "reasoning": "Explain step-by-step why this move makes sense based on graph exploration. Write as if you're naturally discovering paths through the graph.",
    "notebook_update": "Summary of progress from a natural exploration perspective. Write as if you're documenting your discovery process."
}

==User_prompt==

Question: {question}
Current entity: {current_entity} ({entity_name_map.get(current_entity, 'N/A')})

**STATUS (Collected so far):** [{collected_str}]

=== YOUR NOTEBOOK (MEMORY) ===
{notebook if notebook else "Notebook is currently empty."}
==============================

Available neighbor entities:
{neighbors_text}

Reasoning path history: {', '.join(path_history) if path_history else 'None'}

{constraint_msg}

Please decide on the next optimal action to reach the REMAINING Target Answers.

**REMINDER:** If you choose 'finish_search', you MUST include ALL collected answers in the 'entity' field as a list: ["id1", "id2", ...]

Output ONLY valid JSON.

*Figure 5.* The prompt for prompting LLMs to generate agent-based SFT data.

## RL Prompt

**System_prompt=**

You are an expert Graph Exploration Agent.
Your goal is to navigate a knowledge graph to find the correct answer(s) to the user's question.

**CRITICAL INSTRUCTION - METACOGNITIVE SEPARATION:**
- **Simulate a natural, exploratory reasoning process** as if you are discovering answers through graph navigation alone.
- **Example of GOOD reasoning:** "Relation R looks promising because it connects to entities that might be relevant to the question. Let me explore this path."
- **Example of BAD reasoning:** "I know entity X is in the target answers, so I'll mark it."
- **Example of GOOD notebook_update:** "Explored entity X via relation R. Found several connected entities that seem related to the question topic. Continuing to investigate these connections."
- **Example of BAD notebook_update:** "Found entity X which is in the target answers."

**Your Mission:**
1. If the current entity OR ANY of its NEIGHBORS are Target Answers, you MUST use 'mark_answer'. **You can output a LIST of entity IDs to mark multiple answers at once.**
2. If you hit a dead end, Backtrack.
3. If you have found ALL Target Answers, use 'finish_search'.

**You are creating a "gold standard" path.** Make optimal decisions based on the available neighbors to reach the targets, but present your reasoning as natural discovery.
**Efficiency Rule:** If multiple answers are present as neighbors, mark ALL of them in a SINGLE 'mark_answer' action.

You must decide on ONE of four actions:
1. "select_neighbor": Move to a neighbor entity that brings you closer to the Target Answers.
  - For this action, 'entity' must be a list format with length 1: ["entity_id"]
  - **CRITICAL RULE:** The selected entity MUST be one of the entities listed in "Available neighbor entities" section. You CANNOT select an entity that is not in the current neighbors list.
2. "backtrack": Go back if the current branch does not lead to any missing Target Answers.
  - For this action, 'entity' must be a list format with length 1: ["entity_id"]
  - **CRITICAL CONSTRAINT:** You **CANNOT** backtrack to the current node. The `previous_node_id` must be a different node that appears earlier in your path. Backtracking to the current node will not change your position and will cause an infinite loop.
  - **CRITICAL RULE:** Once you backtrack FROM a node, that node is PERMANENTLY EXCLUDED. You CANNOT select it again as a neighbor in future steps. Treat it as if it no longer exists in the graph.
3. "mark_answer": **REQUIRED** if Current Entity OR any Neighbors are in the Target Answers list.
  - **IMPORTANT: For 'mark_answer', 'entity' MUST be a LIST format: ["id1", "id2", ...]**
4. "finish_search": **REQUIRED** if you have found ALL Target Answers in the list.
  - For this action, 'entity' MUST be a LIST format containing ALL collected answers: ["id1", "id2", ...]

**CRITICAL: Your response MUST be valid JSON format ONLY. Output ONLY the JSON object.**
Your response must be in JSON format:
{
    "action": "select_neighbor" | "backtrack" | "mark_answer" | "finish_search",
    "entity": [...],
    "reasoning": "...",
    "notebook_update": "..."
}

**User_prompt=**

Question: {question}

Starting from {num_entities} entity/entities: {start_entities_display}

Reasoning Path Triples: {reasoning_path_triples_json}

**STATUS (Collected so far):** [None yet]

=== YOUR NOTEBOOK (MEMORY) ===
Exploration just started. No steps recorded yet.
=============================

Available neighbor entities:
Neighbor Triples:
No neighbors available (please move to explore).

Reasoning path history: {start_entities_path}

Please decide on the next optimal action to reach the REMAINING Target Answers.

**REMINDER:** If you choose 'finish_search', you MUST include ALL collected answers in the 'entity' field as a list: ["id1", "id2", ...]

Output ONLY valid JSON.

*Figure 6.* The prompt for prompting LLMs in the RL stage.

---

**Comprehensive Evaluation Prompt**

You are an evaluation expert tasked with assessing the comprehensiveness of reasoning steps. You need to evaluate based on the input prompt and output together.

=== Input Information ===
System Prompt (system_prompt):
{system_prompt if system_prompt else 'None'}

User Prompt (user_prompt):
{user_prompt if user_prompt else 'None'}

=== Output Information ===
Reasoning:
{reasoning if reasoning else 'None'}

Notebook Update:
{notebook_update if notebook_update else 'None'}

=== Evaluation Criteria ===
Please evaluate whether the thinking considers all important aspects and is thorough, and provide a score from 0-10:

1. Reasoning Score (0-10 points):
Scoring Guide (0–10):
- 10: Extremely thorough, covering all relevant angles and considerations with depth.
- 8–9: Covers most key aspects clearly and thoughtfully; only minor omissions.
- 6–7: Covers some important aspects, but lacks depth or overlooks notable areas.
- 4–5: Touches on a few relevant points, but overall lacks substance or completeness.
- 1–3: Sparse or shallow treatment of the topic; misses most key aspects.
- 0: No comprehensiveness at all; completely superficial or irrelevant.

2. Notebook Update Score (0-10 points):
Scoring Guide (0–10):
- 10: Extremely thorough, covering all relevant angles and considerations with depth.
- 8–9: Covers most key aspects clearly and thoughtfully; only minor omissions.
- 6–7: Covers some important aspects, but lacks depth or overlooks notable areas.
- 4–5: Touches on a few relevant points, but overall lacks substance or completeness.
- 1–3: Sparse or shallow treatment of the topic; misses most key aspects.
- 0: No comprehensiveness at all; completely superficial or irrelevant.

Please return the result in JSON format:
{{
    "reasoning_score": integer score from 0-10,
    "reasoning_judgment": "Detailed justification (must explain: 1) which important aspects are covered; 2) which important aspects are omitted; 3) the depth of thinking; 4) why this score is given)",
    "notebook_update_score": integer score from 0-10,
    "notebook_update_judgment": "Detailed justification (must explain: 1) which important aspects are covered; 2) which important aspects are omitted; 3) whether it is consistent with reasoning; 4) why this score is given)"
}}

*Figure 7.* The prompt to evaluate comprehensiveness of reasoning steps.

---

**Factual Accuracy Evaluation Prompt**

You are an evaluation expert tasked with assessing the factual accuracy of reasoning steps. You need to evaluate based on the input prompt and output together.

=== Input Information ===
System Prompt (system_prompt):
{system_prompt if system_prompt else 'None'}

User Prompt (user_prompt):
{user_prompt if user_prompt else 'None'}

=== Output Information ===
Reasoning:
{reasoning if reasoning else 'None'}

Notebook Update:
{notebook_update if notebook_update else 'None'}

=== Evaluation Criteria ===
Please evaluate whether the reasoning and notebook_update are based on accurate and verifiable facts, and provide a score from 0-10:

1. Reasoning Score (0-10 points):
Scoring Guide (0–10):
- 10: All facts are accurate and verifiable.
- 8–9: Mostly accurate; only minor factual issues.
- 6–7: Contains some factual inaccuracies or unverified claims.
- 4–5: Several significant factual errors.
- 1–3: Mostly false or misleading.
- 0: Completely fabricated or factually wrong throughout.

2. Notebook Update Score (0-10 points):
Scoring Guide (0–10):
- 10: All facts are accurate and verifiable.
- 8–9: Mostly accurate; only minor factual issues.
- 6–7: Contains some factual inaccuracies or unverified claims.
- 4–5: Several significant factual errors.
- 1–3: Mostly false or misleading.
- 0: Completely fabricated or factually wrong throughout.

Please return the result in JSON format:
{{
    "reasoning_score": integer score from 0-10,
    "reasoning_judgment": "Detailed justification (must explain: 1) which facts are accurate; 2) which facts are inaccurate or unverifiable; 3) why this score is given)",
    "notebook_update_score": integer score from 0-10,
    "notebook_update_judgment": "Detailed justification (must explain: 1) which facts are accurate; 2) which facts are inaccurate or unverifiable; 3) whether it is consistent with reasoning; 4) why this score is given)"
}}

*Figure 8.* The prompt to evaluate faithfulness of reasoning steps.

