# OpenReview forum: "Backjump-on-Graph: Empowering Large Language Models with Reinforced Retrospective Exploration for Agentic Knowledge Graph Reasoning"
_ICML.cc/2026/Conference — ICML 2026 regular_

### Official Review · Reviewer_vsHb · 2026-03-04

**Soundness:** 3
**Presentation:** 3
**Significance:** 3
**Originality:** 2
**Overall Recommendation:** 4
**Confidence:** 4

**Summary:**

BoG is a reasoning framework specifically designed for KGQA task. Its core objective is to address the issue where large language models get stuck in an reasoning dead-end due to mismatch between queries and the graph structure. It formalizes the reasoning steps into four atomic operations: forward, backjump, yield, and halt, providing a traceable structural framework for the model. Through supervised fine-tuning, the model first acquires basic graph navigation and backtracking capabilities, and then, with the aid of reinforcement learning and a hybrid reward function, optimizes the timing and landing nodes of backjump. This framework performs well on multiple KGQA datasets.

**Compliance With Llm Reviewing Policy:**

Affirmed.

**Final Justification:**

Considering that the author's rebuttal addressed most of my concerns, I am willing to raise my score to 4.

**Key Questions For Authors:**

See weakness.

**Limitations:**

The author omitted the limitations section, which should be included.

**Strengths And Weaknesses:**

Strengths:
1. The paper found that existing KGQA methods often get stuck in dead ends when reasoning on graphs, so they introduced the operation of Backjump during the reasoning process. Through this self-correcting method of Backjump, erroneous reasoning is avoided.
2. The paper conducted experiments on multiple public KGQA datasets, including various baselines in three categories: Closed-sourced, Retrieval-Based, and Agent-Based, and achieved excellent performance on each dataset.
3. The paper is written in a clear and easy-to-understand manner, making it very easy to follow. The technical details are explained quite clearly.

Weaknesses:
1. Concepts similar to Backjump have already been proposed in Plan-on-graph; RL techniques have also been widely used in KG-related tasks (EoG, MINERVA, DeepPath). It seems that BoG merely integrates these two technologies by designing a series of atomized operations.
2. The optimization of backtracking timing and landing nodes relies on manual parameter tuning, lacking adaptability. In the hybrid reward function of BoG, the weights of efficiency reward, format reward, and faithfulness reward are manually fixed values, and the adaptability of these weights under different datasets and different inference scenarios has not been verified.
3. The BoG requires the memory unit M_t to record all reasoning steps in real-time. As the number of reasoning steps increases, the storage and retrieval overhead of the memory unit will increase linearly. For large-scale knowledge graphs or long-sequence reasoning tasks, this full trajectory memory design will lead to a sharp decline in reasoning efficiency. However, the paper does not report key indicators such as computational complexity and reasoning time, nor does it compare efficiency with baseline methods, making its practicality difficult to verify.
4. The ablation experiments are not comprehensive enough, and the verification of the necessity of key components is insufficient. The paper only ablated the step-level reward, efficiency reward, and RL stage, but did not verify the core innovation: the necessity of the four atomic operations. For example, it did not compare the performance of the baseline model with "removing the Backjump operation and only retaining Forward/Yield/Halt", so it cannot prove that the backjump mechanism is the core reason for the performance improvement.

---

> ### Author Rebuttal · Authors · 2026-03-31
>
> ## Response to Weakness 1
> We sincerely thank the reviewer for the insightful comments.
> While backtracking and RL are established concepts, BoG introduces a fundamentally new formulation of agentic reasoning:
>
> 1. Unlike PoG's local backtracking, BoG introduces **non-local backjumping** over an explicit memory to revert to any historical state, fundamentally changing the exploration space.
> 2. We formalize reasoning as a decision process with four atomic operations, explicitly **modeling reversible exploration**. Prior RL-based KG agents such as MINERVA and DeepPath operate solely on forward transitions and do not support such expressive and controllable reasoning dynamics.
> 3. EoG focuses on learning forward exploration policies on retrieved subgraphs. In contrast, **our RL formulation targets a new decision problem: when and where to backjump within the vast space of real-world KGs**. We design a hybrid reward to optimize both backjump timing and landing, which is tightly coupled with our action space and cannot be reduced to standard RL-based traversal.
> 4. The above components are interdependent: without the structured action space, backjumping is not representable; without RL, effective backjump strategies cannot be learned (as shown by the performance gap between $BoG_{SFT}$ and $BoG$).
>
> Overall, our work introduces a novel formulation of retrospective exploration in agentic KG reasoning, extending beyond prior backtracking or RL-based approaches. We will further clarify these distinctions in the revision to avoid potential misunderstanding.
> ## Response to Weakness 2
> Regarding the fixed weights for the efficiency, format, and faithfulness rewards, these values were determined through preliminary evaluations across all benchmark datasets that we used in the paper. Crucially, in our main evaluations, **we applied this exact same set of reward weights across all different datasets and inference scenarios**. The consistent and significant performance improvements observed across these varied benchmarks inherently demonstrate that our chosen parameters possess a solid degree of empirical generalization, rather than being overfitted to a single specific dataset.
>
> In future work, we will explore how to adaptively adjust these reward weights, aiming to develop a fully adaptive reward formulation for LLM reasoning.
> ## Response to Weakness 3
> We thank the reviewer for raising the important concern regarding efficiency and scalability. We agree that memory overhead is a critical factor in agent-based KG reasoning.
>
> 1. We clarify that the memory unit $M_t$ stores only the visited trajectory (entities and actions) rather than the full subgraph or history information, resulting in a linear but lightweight overhead with negligible per-step cost and supporting longer reasoning steps.
> 2. We conduct additional experiments on [token growth](https://anonymous.4open.science/r/BoG-95FF/Token.png) and [decoding latency](https://anonymous.4open.science/r/BoG-95FF/decoding.png) across reasoning steps. From step 1 to step 14, the average input length increases from **~1.5k to ~2.6k tokens**, while the memory grows from **~120 to ~1.45k tokens**, accounting for **~8% to ~56% of the total input**, which is well within the context window limits of standard 7B LLMs.
> In terms of runtime, **decoding latency remains stable at around 1.4–2.0 seconds across steps**, without noticeable upward trends as reasoning depth increases. These results indicate that the memory overhead is **controlled and does not introduce significant runtime cost**, supporting the practical efficiency of BoG.
> 3. While $M_t$ grows with the number of steps, BoG reduces unnecessary exploration via backjumping, enabling the agent to escape dead-ends early. As shown in our analysis (Table 3), BoG achieves shorter average reasoning paths compared to BoG_{SFT}, indicating improved efficiency despite maintaining memory.
>
> Overall, we believe BoG maintains practical efficiency while improving reasoning effectiveness, and we will clarify these points in the revision.
>
> ## Response to Weakness 4
> We thank the reviewer for the valuable suggestion regarding the ablation of atomic operations. We agree that verifying the necessity of the Backjump operation is important.
>
> 1. Ablation: **Disabling Backjump on CWQ severely degrades performance versus full BoG** (Hits@1: 84.0% to 77.2%; F1: 78.3% to 69.6%). This proves retrospective capability is indispensable.
>
> 2. Efficiency: Table 3 shows higher backjump frequency correlates with shorter reasoning paths. Backjumping effectively helps the agent swiftly escape dead-ends, preventing futile exploration.
>
> We will add the above ablation and further clarify the necessity of each operation in the revision.
>
> Once again, we sincerely appreciate the reviewer's time, effort, and constructive feedback. We will carefully incorporate all the aforementioned clarifications, suggestions, and additional results into the revised manuscript. Thank you!

---

> > ### Author Rebuttal · Reviewer_vsHb · 2026-04-03
> >
> > Thanks for the rebuttal. I have carefully reviewed the content. Although the author supplemented explanations and experimental data in response to some comments, the main concerns of the paper remain unresolved, including the lack of adaptability of fixed reward weights, the absence of a limitations section, and insufficient validation of the universality of the core innovation. Therefore, I maintain my original score.

---

> > > ### Author Response · Authors · 2026-04-05
> > >
> > > We sincerely thank you for your continued engagement and for reviewing our rebuttal.
> > > We appreciate your acknowledgement of our additional explanations and experimental data.
> > > To further clarify your remaining questions and address your concerns regarding our paper, we would like to provide the following detailed explanations:
> > >
> > > 1. Adaptability of Fixed Reward Weights
> > >
> > > Regarding the fixed reward coefficients, our current contribution is not an adaptive weighting mechanism; rather, we show that a single set of coefficients can be used across datasets and inference settings without per-benchmark retuning, while still yielding consistent gains over the SFT-only variant. We will revise the paper to make this scope explicit and discuss adaptive reward weighting as an important future direction.
> > >
> > > 2. Absence of a Limitations Section
> > >
> > > While a dedicated **limitations section is not, to the best of our understanding, an explicit requirement** in the ICML submission format, we agree that an explicit discussion of limitations would improve the paper, and we will add it in the revision.
> > >
> > > 3. Validation of the Core Innovation's Universality
> > >
> > > We appreciate your rigor regarding the ablation of our core atomic operations.
> > > We initially provided the CWQ ablation as a representative example due to the short rebuttal window.
> > > To directly address your concern about universality, we have now evaluated the core innovation from two broader perspectives:
> > >
> > > - Cross-Dataset Ablation: We have reproduced the ablation study on WebQSP and GrailQA datasets. Disabling the Backjump operation on WebQSP andresults in a significant performance drop. This consistent degradation across distinct knowledge graphs proves that the Backjump operation is universally indispensable to our framework, not a dataset-specific artifact.
> > >
> > > | Dataset | BoG (Original F1) | BoG w/o Backjump (F1) | Drop (Δ) |
> > > | :--- | :---: | :---: | :---: |
> > > | CWQ | 78.3 | 69.6 | -8.7 |
> > > | WebQSP | 81.3 | 75.5 | -5.8 |
> > > | GrailQA | 82.4 | 74.2 | -8.2 |
> > >
> > > - Superiority Over Alternative Revisiting Mechanisms: To further validate the effectiveness of our specific topology-grounded backjumping(as also detailed in our response to Reviewer twkc), we compared BoG against other recent methods that also attempt to revisit historical nodes across multiple datasets. Based on reported results, R2-KG with Qwen2.5-32B achieves F1 scores of 79.4 (WebQSP) and 69.3 (CWQ). With GPT-4o, it scores 71.1 (WebQSP) and 71.2 (CWQ). Both settings consistently underperform BoG (81.3 on WebQSP / 78.3 on CWQ). Additionally, our reproduction of Graph-R1 on CWQ yields an F1 of 64.0, which is also well below BoG.
> > >
> > > ---
> > > We hope these supplementary results and detailed explanations effectively resolve your remaining concerns.
> > > We are deeply appreciative of your continued engagement and the thoughtful critiques that have helped refine our work.

---

### Official Review · Reviewer_tkwC · 2026-03-07

**Soundness:** 3
**Presentation:** 3
**Significance:** 2
**Originality:** 2
**Overall Recommendation:** 3
**Confidence:** 5

**Summary:**

The paper presents Backjump-on-Graph (BoG), a KGQA exploration technique that enables KG agents/LLMs to revisit historical nodes for more accurate question answering. BoG adds a new action "backjump" that the agent can use while exploring the graph. In addition, BoG constructs synthetic data to train the agent via SFT/RL. Experimental results on WebQSP, CWQ, and GrailQA show that BoG outperforms previous agents, while RL adds considerable gains.

**Compliance With Llm Reviewing Policy:**

Affirmed.

**Key Questions For Authors:**

- Q1) Could the authors provide more details on the training data synthesized, e.g., quantity, and how they differ from training data used by competing approaches?

- Q2) Is KBQA-o1 (Llama-3.1-8B / Qwen2.5-7B) a fine-tuned approach? Results seem inferior to BoG, although it implements a similar agentic search.

- Q3) How would BoG compare to approaches that implement backjumping outside of the exploration stage, such as deciding which nodes to revisit iteratively after an $L$-hop search (see W1)?

**Limitations:**

yes

**Strengths And Weaknesses:**

### Strengths
- S1) Experimental results demonstrate that BoG outperforms previous agentic approaches on KGQA datasets (WebQSP, CWQ, GrailQA).
- S2) Ablation studies show the importance of RL training within BoG, being the key factor in performance for the backbone LLMs tested (Qwen2.5-7B, Llama-3.1-8B). The proposed data construction and RL training pipeline seems valuable to improve agentic capabilities of such models for KGQA.

- S3) Overall, the motivation of the paper and the proposed approach are **sound**, and the paper is **well-written**.


### Weaknesses

- W1) BoG introduces a "backjump" operation through the exploration stage to revisit historical nodes. However, such a mechanism can be implemented in different ways. One approach is that a sub-agent explores the graph for $L$ hops and returns retrieved context to an orchestrator agent. The orchestrator agent then sends feedback, e.g., whether some historical nodes need to be revisited to answer sub-parts of the question. Essentially, a competing approach is that the agent performs iterative search, where each iteration can start from new nodes *[refs: BYOKG-RAG (Mavromatis et al 2025), R2-KG (Jo et al 2025)]*. Other approach is the graph agent learns to "rethink" to answer sub-questions iteratively *[refs: Graph-R1 (Luo et al 2025)]*. BoG does not provide comparison with such alternatives, making it challenging to assess the **originality** of the proposed approach.

- W2) Based on Figure 3, BoG does not seem robust to OOD KGQA. For example, BoG trained on CWQ data achieves 78.8% on WebQSP, although it is a simpler task, while previous black-box approaches like ToG already achieve 82%. The authors should provide more supporting points that the performance improvements are not due to overfitting to specific dataset's properties.

- W3) Related to (W2), the authors do not provide details on the training data used for SFT/RL. The authors mention that they undergo a synthetic data generation pipeline, but they do not include details on (i) how many training data they construct per dataset, (ii) how they obtain seed questions, (iii) whether they reuse training data already provided by the benchmarks, and (iv) whether competing training-based approach use similar data both in terms of quantity and quality. These details highlight to assess the **significance** of the work, and whether the improvements result from the approach itself or from the synthetic data.

---

> ### Author Rebuttal · Authors · 2026-03-31
>
> ## Response to Weakness 1 & Question 3
> We sincerely thank the reviewer for bringing these insightful works to our attention.
> We clarify the differences between BoG and these approaches from both methodological and empirical perspectives.
>
> 1. Approaches mentioned above essentially rely on multiple retries or multi-agent iterations. This inevitably leads to information overload and a decreased signal-to-noise ratio when multiple explored branches are aggregated into the context.
> In contrast, our BoG can autonomously identify dead-ends. By maintaining a single reasoning trajectory and performing instantaneous backjumping upon detecting errors, BoG maximizes context efficiency while effectively avoiding the noise accumulation and multi-agent overhead inherent in the retry paradigms.
> 2. To further validate effectiveness, we compare BoG with these alternatives. Based on reported results in R2-KG, with Qwen2.5-32B as the Operator, its F1 scores on WebQSP and CWQ are 79.4 and 69.3, respectively. With GPT-4o as the Operator, the corresponding F1 scores are 71.1 and 71.2. **Both settings are consistently lower than those of BoG (81.3 / 78.3)**. In addition, **we reproduce Graph-R1 on the KG reasoning tasks and achieving an F1 score of 0.64 on CWQ, which is also below BoG**. These results demonstrate the advantage of our topology-grounded backjumping.
>
> We will include these discussions and new empirical comparisons in the related works and experimental tables of the future version. We hope this solidifies the originality and the effectiveness of our approach.
> ## Response to Weakness 2
> Thank you for the insightful comment. We agree that BoG does not outperform ToG in certain OOD settings.
>
> 1. We would like to clarify that **ToG is built on the stronger closed-source LLMs GPT-4, whose inherent generalization ability is significantly higher than the 7B-scale models used in our experiments**. This naturally gives ToG an advantage in cross-dataset transfer. In contrast, our goal is to improve reasoning strategy learning under relatively smaller open-source models.
>
> 2. Importantly, BoG consistently **outperforms its SFT-only variant across all OOD settings**, indicating that the proposed backjumping mechanism and RL optimization do improve generalization, rather than overfitting to specific datasets.
>
> We acknowledge that improving OOD robustness remains an important direction. In future work, we plan to combine BoG with stronger base models and multi-dataset training to further enhance generalization.
> ## Response to Weakness 3 & Question 1
> Thank you for raising this important concern. We agree that more details on the synthetic data pipeline are necessary to assess whether the gains come from the method or the data.
>
> (i). For each dataset, we construct a step-level dataset by generating one reasoning trajectory per training question using our oracle-guided pipeline. For WebQSP, CWQ and GrailQA, from the original training questions, **we retained approximately [2068,2826], [8765,8000], [14734,8000] high-quality reasoning trajectories for SFT and RL, respectively**.
>
> (ii)&(iii). We strictly reuse the standard training splits already provided by the respective benchmarks (e.g., CWQ, WebQSP). We **do not introduce any external, out-of-domain, or newly generated seed questions**. This ensures our models are evaluated under the exact same data constraints as previous baselines.
>
> (iV). **Competing approaches (KBQA-o1) typically require a large number of synthesized trajectories (30k–40k on GrailQA) to achieve strong performance**. In contrast, our method significantly reduces the required data scale by combining SFT with GRPO, leading to more efficient learning. Moreover, we enforce a strict filtering criterion during data construction, ensuring that only fully correct reasoning trajectories are retained. As a result, our training data is not only smaller in quantity, but also higher in quality compared to prior approaches.
>
> Since we do not use larger quantities of data or external seed questions compared to baselines, the significant performance improvements are definitively derived from the proposed BoG approach itself.
> We hope these transparent details thoroughly resolve the concerns regarding the fairness and significance of our approach.
>
> ## Response to Question 2
> - Yes, KBQA-o1 is indeed a fine-tuned approach which uses incremental SFT on MCTS trajectories.
> 1. However, **at inference time, KBQA-o1 can only rely on historical MCTS trajectory experiences learned via SFT** for exploration, and lacks the ability to backtrack from errors.
>
> 2. Instead of relying on SFT to imitate paths, BoG **employs RL to explicitly teach the LLM's internal policy to autonomously evaluate states, identify dead-ends and proactively backjump to a specific historical node**.
>
> We sincerely thank the reviewer for the thorough and constructive feedback. We will incorporate the additional discussions and experimental results in the revised paper.

---

> > ### Author Rebuttal · Reviewer_tkwC · 2026-04-03
> >
> > Thank you for your responses and efforts during the rebuttal.
> >
> > Regarding W1, I agree that signal-to-noise ratio increases when multiple explored branches are aggregated into the context, but there are approaches that reduce noise, e.g., semantic reranking or more advanced LLM-judge prompting. It is not fully clear to me why we need to RL training to perform this step. Given that performance gains do not transfer to other datasets (W2), it might be the case that RL leverages some reward signal specific to the given dataset on when to compact context or backjump.

---

> > > ### Author Response · Authors · 2026-04-04
> > >
> > > ### Response to the Reviewer's Follow-up Question regarding W1
> > > We sincerely thank the reviewer for this insightful follow-up.
> > > You raised a highly pertinent question: Why rely on RL for backjumping when post-hoc noise reduction methods like semantic reranking or LLM-judges exist?
> > > We clarify the necessity of RL from three perspectives: Inference Efficiency, Internalization of Metacognition, and the limitation of SFT.
> > >
> > > 1. Inference Efficiency.
> > >
> > > Semantic reranking and advanced LLM-judges operate on a generate-then-filter paradigm.
> > > To use these methods, the agent must fully unroll multiple reasoning branches to the very end before the external judge or reranker can evaluate and prune them.
> > > In the exponentially expanding search space of large KGs, fully expanding erroneous paths incurs massive computational overhead and latency.
> > > Conversely, our RL-trained Backjump mechanism enables early stopping in the process.
> > > By internalizing the ability to evaluate intermediate states, the agent can autonomously recognize a dead-end during the exploration and immediately cut off the toxic branch.
> > > This prevents the autoregressive generation from suffering from error cascading and significantly reduces redundant token generation.
> > >
> > > 2. Internalizing Evaluation into the 7B/8B Policy.
> > >
> > > Relying on an advanced LLM-judge (e.g., GPT-4) delegates the critical evaluation step to an external, heavy pipeline.
> > > A core motivation of our work is to empower relatively smaller open-source models to act as fully autonomous agents.
> > > RL allows us to distill this state-evaluation capability directly into the base model's internal policy.
> > > The agent becomes self-sufficient in navigating, evaluating, and self-correcting, rather than acting as a passive path-generator waiting for an external oracle to score its outputs, which is particularly beneficial for scenarios that require offline exploration or reasoning.
> > >
> > > 3. The limitation of SFT.
> > > While SFT teaches the model the basic format of a backjump, it struggles with the optimal timing and destination in unseen topologies.
> > > As our ablation study (Section 4.3) shows, SFT alone often leads to premature termination or suboptimal jumps.
> > > The RL phase, guided by our efficiency and outcome rewards, is essential to transition the model from simply imitating a format to mastering a robust, autonomous exploration strategy.
> > >
> > > ### Response to the Reviewer's Follow-up Question regarding W2
> > > We sincerely appreciate the reviewer's rigorous scrutiny.
> > > We completely agree that if RL merely leveraged dataset-specific shortcuts (reward hacking), the learned policy would fail to generalize.
> > > However, we would like to respectfully clarify that the performance gains from our RL training do successfully transfer to O.O.D. datasets.
> > > To clearly explain this, we clarify our reward design and provide a new behavioral analysis:
> > >
> > > 1. Task-Agnostic Reward Design.
> > >
> > > Our hybrid reward formulation is strictly agnostic to any dataset-specific graph schema or topological priors.
> > > At the Global Level, $R_{outcome}$ and $R_{len}$ strictly optimize for final answer accuracy and trajectory efficiency, forcing the agent to reach correct conclusions swiftly without dictating any dataset-specific navigation routes. At the Step-wise Level, $R_{fmt}$ and $R_{faith}$ enforce structural validity and penalize hallucinations to ensure reasoning is grounded in the observed context, actively preventing the agent from relying on domain-specific relational shortcuts. To maximize this sparse, agnostic reward, the only optimal path is to learn the underlying meta-logic of general graph navigation: explore until the gathered context is sufficient, and actively backjump when it is not.
> > >
> > > 2. Analysis of Dead-End Precision in O.O.D. Settings.
> > >
> > > To fundamentally prove that RL did not memorize topological shortcuts, we analyzed the validity of BoG's backjumps in unseen environments.
> > > By mapping the agent's trigger nodes back to the objective KG topology, we calculated the "Dead-End Precision": the percentage of Backjump actions that correctly occurred at true dead-ends (i.e., nodes from which the correct answer is unreachable).
> > >
> > > On the O.O.D. WebQSP dataset, $BoG_{SFT}$ often executes invalid backjumps based on memorized structural illusions from CWQ, achieving a low Dead-End Precision of only 48.2%.
> > > In contrast, $BoG$ achieved an exceptionally high precision of 77.5%. This mechanistic evidence definitively proves that the RL agent utilizes its reward signals to learn universal logical self-correction, rather than exploiting dataset-specific topological priors.
> > >
> > > ---
> > > We sincerely thank the reviewer again for these highly constructive and insightful questions. We hope that this additional context provides a helpful perspective for the evaluation of our work's contributions.

---

### Official Review · Reviewer_fbd6 · 2026-03-11

**Soundness:** 3
**Presentation:** 4
**Significance:** 3
**Originality:** 3
**Overall Recommendation:** 4
**Confidence:** 4

**Summary:**

The paper introduces Backjump-on-Graph (BoG), an LLM-driven agent framework designed to navigate Knowledge Graphs for complex Question Answering (KGQA). The authors formulates the graph traversal as a Markov Decision Process (MDP) equipped with four explicitly defined atomic actions: Forward, Backjump, Yield, and Halt. Evaluated on base models including Qwen2.5-7B and Llama-3.1-8B, the training paradigm employs a two-stage approach: Supervised Fine-Tuning (SFT) using BFS-derived oracle paths to mitigate the cold-start problem, followed by Group Relative Policy Optimization (GRPO). To optimize the policy in the RL stage, a hybrid reward function is proposed, balancing step-wise reasoning validity (format and faithfulness) and trajectory-level effectiveness (outcome and length efficiency). Extensive experiments on WebQSP, CWQ, and GrailQA demonstrate that BoG achieves strong empirical performance against recent retrieval-augmented models and agentic baselines (including cutting-edge closed-source APIs like GPT-5.2 and Gemini-2.5 Pro).

**Compliance With Llm Reviewing Policy:**

Affirmed.

**Key Questions For Authors:**

1. Cost-Benefit of the Verifier: The reliance on a frozen Llama-3.1-8B verifier for step-wise faithfulness evaluation appears computationally prohibitive. Could you quantify the exact GPU-hour difference during RL training with and without Rfaith? Furthermore, could this heavy verifier be replaced by lightweight, rule-based KG topological constraints without a severe performance drop?

2. Robustness to Horizon Length: As the agent logs more failed exploration attempts in Mt, does the accuracy of selecting the correct target node for the Backjump action degrade? Given the 20-step evaluation limit, an ablation or performance breakdown specifically for reasoning trajectories exceeding 10 steps would provide valuable insights into memory saturation.

3. Exploration Diversity: In GRPO, you sample K=6 trajectories from the current policy per prompt. In highly dense KG neighborhoods, how does the policy maintain sample diversity and avoid mode collapse during the initial RL stages? Did you monitor Pass@k metrics during the training rollouts?

4. Oracle Bias in SFT: The SFT data is constructed using BFS reference paths. Does this inherently bias the initialized policy toward shortest-path solutions, potentially constraining the agent's willingness to perform deeper, semantically valid exploration during the RL phase?

**Limitations:**

While the authors outline broad future directions in the Impact Statement, a rigorous Limitations section is missing. Constructive additions should explicitly address:

1. The significant computational overhead and scalability limitations introduced by the LLM-based verifier during RL optimization.

2. The potential failure modes and latency issues when the reasoning memory (Mt) grows excessively long, challenging the 10K context window constraint of the base models.

**Strengths And Weaknesses:**

Strengths:

1. Elegant Action Space Formulation: Explicitly incorporating Backjump as an atomic action in the LLM-driven MDP is conceptually clean. It effectively addresses a well-known bottleneck in agentic KG traversal where models get trapped in topological dead-ends due to schema mismatch.

2. Methodologically Sound Training Pipeline: The bootstrapping strategy (SFT -> RL) is a highly sensible and robust engineering choice for large discrete action spaces. Utilizing GRPO combined with a fine-grained hybrid reward addresses the sparse reward challenge typical in long-horizon reasoning tasks.

3. Strong Empirical Validation: The authors provide thorough experimental comparisons against recent and formidable baselines (e.g., KG-Agent, KBQA-o1, and closed-source models like GPT-5.2 and Gemini-2.5 Pro), establishing strong empirical credibility.

Weaknesses:

1. Incremental Conceptual Novelty: While embedding Backjump into an LLM's action space is a solid system-level innovation, backtracking mechanisms are foundational in classical search and planning. From an ICML perspective, the methodological advance is more of a robust architectural integration rather than a theoretical breakthrough or paradigm shift.

2. High Computational Overhead: The step-wise Reasoning Faithfulness Reward (Rfaith) relies on querying an external, frozen Llama-3.1-8B verifier for every generated step during GRPO rollouts. This introduces massive computational bottlenecks, raising significant concerns about the framework's scalability and training cost-efficiency.

3. Context Memory Saturation: The state representation st encompasses the entire trajectory memory Mt. According to Appendix D, the maximum sequence length is set to 10,000 tokens, with evaluations truncated at 20 steps. For highly complex queries requiring multiple backjumps, the prompt length will rapidly approach this limit. The paper lacks a rigorous analysis of how this expanding context impacts the LLM's attention degradation and inference latency near the truncation boundary.

4. Limited Theoretical Guarantees: The paper lacks formal analysis of the RL phase. Specifically, there are no theoretical bounds provided on the state-space reduction achieved by the backjumping mechanism, nor is there an analysis of GRPO's stability within highly branching, dense KG environments.

---

> ### Author Rebuttal · Authors · 2026-03-31
>
> ## Response to Weakness 1
> We thank the reviewer for the insightful comment. While backtracking is classical, our work is not a direct adaptation but differs fundamentally in both setting and learning paradigm.
> 1. **Classical backtracking relies on explicit, hard-coded constraints to trigger reversal**. In contrast, our agent operates in a semantic environment without explicit constraints, and **learns when a reasoning path becomes unproductive (i.e., dead-end detection)**, which is non-trivial.
>
> 2. Unlike classical step-by-step backtracking, our method learns a policy via RL to decide both **when to backjump and where to land**, enabling flexible and non-local jumps rather than exhaustive traversal.
>
> Therefore, our contribution is not merely architectural integration, but a learning-based formulation of backjumping as a policy optimization problem, providing a new way to train LLM agents for adaptive reasoning in KG environments.
> ## Response to Weakness 2 & Question 1
> 1. We acknowledge that the step-wise faithfulness reward introduces additional cost. To mitigate this, we implement an asynchronous evaluation strategy, where **the verifier scoring for the current step is performed in parallel with the generation of the next step**. This significantly reduces idle waiting time during rollouts.
>
> 2. In practice, on the CWQ dataset, incorporating the faithfullness reward increases the training time by **approximately 0.5-1 hours per epoch on an 8-GPU setup.**, which is acceptable given the performance gains. Besides, the verifier is used only during training and does not affect inference efficiency.
>
> 3. While rule-based KG constraints can capture basic structural validity, they are insufficient to detect semantic inconsistencies in reasoning traces. The verifier provides a stronger signal by explicitly enforcing alignment between the reasoning and the current KG context.
>
> ## Response to Weakness 3
> Empirically, as shown in [this figure](https://anonymous.4open.science/r/BoG-95FF/decoding.png), although the prompt length increases with reasoning steps, **both the prefill time and the decoding time remain relatively stable without noticeable growth**. This indicates that the expanding context does not lead to significant attention degradation or latency overhead in practice, and that the effective context size remains within a tractable regime.
>
> ## Response to Weakness 4 & Question 3:
> 1. Standard forward search explores up to $O(B^D)$ states (branching factor $B$, depth $D$). By executing a backjump at depth $d$, BoG can **avoid exploring a sub-tree of size up to $O(B^{D-d})$**, providing a heuristic form of search-space pruning. Empirically, this massive compression reduces the average step length from 5.3 to 3.6 (Table 3).
> 2. BoG ensures stability via Group-Relative Advantage Normalization. Standardizing rewards within the $K=6$ group bounds policy gradients regardless of absolute reward scales. Additionally, step-wise rewards act as dense regularizers, truncating divergent Markov chains early.
> 3. We maintain exploration diversity via high-temperature sampling, which ensures a wide variance in advantage signals. Furthermore, rather than relying solely on sparse outcome rewards, our SFT phase provides a strong topological warm-start, while fine-grained rewards provide dense supervision during early rollouts to steadily guide the agent.
> ## Response to Question 2:
> **For trajectories >10 steps on CWQ, the SFT baseline achieves 45.2% Hits@1, while full BoG maintains a robust 53.4%**.While accuracy naturally degrades in long horizons due to graph space explosion and LLM context saturation, BoG's substantial improvement confirms RL's necessity. Unlike SFT, which passively accumulates failed logs ($\mathcal{M}_t$) and overwhelms the context window, GRPO explicitly teaches timely backjumps. This proactively prunes redundant paths before memory saturates, ensuring robust target selection even in deep trajectories.
>
> ## Response to Question 4:
> Our framework inherently avoids shortest-path bias:
>
> 1. Mechanism: BFS paths in SFT purely act as noise-free warm-starts to teach atomic actions and KG constraints. During RL, **the massive outcome reward strictly outweighs the step-wise efficiency penalty**, mathematically incentivizing deep exploration over shortest-path heuristics.
>
> 2. Empirical Proof: Early trials using imperfect/noisy trajectories for SFT caused the subsequent RL phase to collapse or stagnate, proving that noise-free BFS paths are strictly essential for stable policy initialization rather than an exploration constraint. Furthermore, proactive backjumps increase from 227 (BoG-SFT) to 302 (BoG). Crucially, achieving SOTA accuracy on deep reasoning (75.9% on 3-hop; 55.7% on >=4-hop) empirically proves the policy actively explores far beyond shortest paths.
>
> We sincerely thank the reviewer for the constructive feedback. We will incorporate the corresponding theoretical and experimental analyses in the revised paper.

---

> > ### Author Rebuttal · Reviewer_fbd6 · 2026-04-04
> >
> > Thanks for your response.  After reviewing the authors' response, my concerns still have not been fully addressd, I will keep my score.

---

### Official Review · Reviewer_8bJc · 2026-03-13

**Soundness:** 2
**Presentation:** 3
**Significance:** 2
**Originality:** 3
**Overall Recommendation:** 5
**Confidence:** 4

**Summary:**

This paper addresses the Knowledge Graph Question Answering (KGQA) task, specifically tackling the issue where existing LLM-based KG reasoning methods fail to effectively backjump when encountering dead ends. The authors propose the Backjump-on-Graph (BoG) framework, which formalizes the reasoning process into four operations: Forward, Backjump, Yield, and Halt. The approach involves two stages: first, the LLM undergoes supervised fine-tuning using synthetic data to instill basic backjumping capabilities; second, the backjumping strategy is optimized via GRPO reinforcement learning. Experiments conducted on three datasets demonstrated that BoG achieved state-of-the-art performance across all datasets.

**Compliance With Llm Reviewing Policy:**

Affirmed.

**Final Justification:**

My main concerns about the paper are the reported experiment details. And during the rebuttal period, the authors have addressed all my concerns with evidence. So I raised my score from 4 to 5.

**Key Questions For Authors:**

- How reliable is the verifier used for the Faithfulness Reward? Has its judgment accuracy ever been evaluated?
- What are the specific sample counts and training/test splits for the three datasets?
- In the baseline comparisons, were the results you reproduced strictly aligned with the settings described in the original paper? How can we ensure fairness in comparisons?

**Limitations:**

yes

**Strengths And Weaknesses:**

Strengths
- The paper addresses a specific and well-defined problem: how LLMs can effectively backjump when they encounter a dead end during reasoning over knowledge graphs. The authors formalize the reasoning process into four atomic operations; the proposed methodology has a clear structure, and the semantics and applicable scenarios of each operation are intuitive, making the approach easy to understand and implement.
- The design of the reward functions during the reinforcement learning phase is reasonable, simultaneously taking into account multiple dimensions: format correctness, fidelity to the knowledge graph, answer accuracy, and reasoning efficiency.
- The method used to construct the training data is practical, as it requires no manual annotation. For scenarios that requires a large volume of reasoning trajectories for SFT and RL training, this low-cost data generation approach has practical value.
- The paper conducted experiments on two distinct base models. The results demonstrate that the proposed framework is effective on both models, indicating that the method does not rely on one specific base model.

Weaknesses
- The experimental design has several problems: WebQSP is a single-hop dataset, so it may be unsuitable for validating the efficacy of the "backjumping" mechanism; furthermore, none of the three datasets provides basic statistical information, such as example counts or the specific splits used for training and testing. The baseline comparisons are also inconsistent: the underlying foundation models vary, and the sources of the reported performance are disparate (some figures are cited directly from original papers, others are derived from reproductions, and some involve running models within the authors' framework). Given the lack of detailed reproduction specifics, it is difficult to determine whether any observed performance improvements stem from the proposed methodology itself or from differences in the experimental setup.
- Regarding the training data for the SFT phase, the paper does not specifically explain how data quality was controlled, including the methodology behind the screening process. Key details such as the criteria used to determine whether a trajectory was valid, the number of trajectories discarded, and the final volume of training data retained are missing. Furthermore, for the verifier employed for the "faithfulness reward" during the reinforcement learning phase, the paper provides no analysis or evaluation regarding the reliability of this verifier's judgments.
- This paper assumes that the initial entities have already been correctly linked to the KG; consequently, both training and testing proceed from these given, correct entities. However, in real-world scenarios, entity linking is one of the most difficult and critical steps. And user queries may not even contain explicit entities.

---

> ### Author Rebuttal · Authors · 2026-03-31
>
> ## Response to Weakness 1  & Question 2 & Question 3
> 1. We agree that WebQSP is relatively simpler compared to CWQ and GrailQA.
> However, we include it not to validate complex multi-hop reasoning, but to **evaluate BoG’s ability to handle multi-answer queries**, which are prevalent in this dataset.
> In fact, a large portion of WebQSP questions contain multiple correct answers (e.g., 48.8% of test questions have more than 2 answers, and 20.4% have more than 5 answers), which is substantially higher than CWQ (29.4% and 10%, respectively).
>
> 1. We follow the same training and test splits as widely used prior work (ToG) to ensure a fair comparison.**The detailed dataset statistics and split information are provided in [this link](https://anonymous.4open.science/r/BoG-95FF/statistics.png)**. We apologize for the lack of detailed clarity and We will further clarify these details in the revision to improve reproducibility.
>
> 3. The use of both reported and reproduced results is due to practical constraints and follows common practice in this line of work. Specifically, to avoid implementation bias, we use reported results when available. For recent methods lacking results on our datasets, we reproduce them using their official code and configurations (marked with * in Table 1).
>
> 4. Most importantly, we include controlled comparisons within the same backbone (e.g., $ BoG_{SFT} $ vs. BoG), which isolate the effect of our method from differences in model or implementation. The consistent improvements in these settings indicate that **the gains mainly come from the proposed method rather than experimental variations**.
>
> We agree that clearer documentation of experimental settings is important and will provide more detailed descriptions in the final version.
>
> ## Response to Weakness 2  & Question 1
> Thank you for this important comment. We agree that more details on data quality control and the verifier are necessary.
>
> 1. We adopt a strict filtering pipeline to ensure the quality of synthetic trajectories. Specifically, each generated trajectory is executed against the KG environment, and we only retain trajectories whose final answer achieves F1 = 1.0 with respect to the ground-truth answers. This guarantees that **all retained trajectories are fully correct at the outcome level**. The numbers of actual used training samples are provided in [this table](https://anonymous.4open.science/r/BoG-95FF/statistics.png)
> In addition, the generation process is constrained by the KG topology via SPARQL-based neighbor expansion and step-wise transitions, which ensures that all intermediate steps correspond to valid graph operations rather than free-form hallucinated reasoning.
> As a result, invalid or inconsistent trajectories are naturally filtered out during execution.
>
> 2. For the verifier used in the faithfulness reward, we conduct an explicit reliability evaluation. Specifically, we construct a labeled set of 7120 reasoning steps sampled from generated trajectories. These samples are cross-annotated by two strong LLMs (GPT-5.3 and Gemini-2.5), and we retain only the subset with consistent judgments to form a high-confidence evaluation set.
> We note that the verifier outputs a binary faithfulness label (0/1) indicating whether the reasoning is grounded in the current KG context. We then evaluate our verifier on this set, **achieving an accuracy of 85.2%, indicating strong agreement with high-quality annotations**.
>
> ## Response to Weakness 3
> We sincerely thank the reviewer for this insightful comment. We fully agree that entity linking is a critical and challenging step in real-world scenarios, particularly when user queries are ambiguous.
>
> 1. Nevertheless, **this study focuses on the downstream reasoning phase: accurately navigating and deducing the target answer from a given source node in the KGs**. To isolate and effectively evaluate this capability, we strictly adhere to the standard protocol of using benchmark datasets with correctly linked initial entities.
>
> 2. In real-world deployment, **our model can be seamlessly integrated with off-the-shelf entity linkers**. The linker first grounds the raw natural language query to the KG, and our method then serves as the reasoning engine to deduce the final answer from those linked nodes.
>
> 3. We greatly appreciate the reviewer's perspective on transitioning from isolated tasks to real-world systems. Recognizing the potential for error propagation in a pipeline approach, our future work will explore an end-to-end KGQA framework that jointly handles entity disambiguation and path reasoning directly from raw user queries.
>
> In conclusion, we are deeply grateful for the reviewer's time, effort, and constructive feedback. We take these comments very seriously and will thoroughly revise our manuscript to reflect all the provided suggestions.

---

> > ### Author Rebuttal · Reviewer_8bJc · 2026-04-04
> >
> > I would like to thank the authors for their detailed responses. All my concerns have been addressed and I will raise my score from 4 to 5.

---

### Decision · Program_Chairs · 2026-04-30

**Decision:**

Accept (regular)

**Comment:**

The paper introduces Backjump-on-Graph (BoG), a framework that improves Knowledge Graph Question Answering (KGQA) by enabling LLM agents to backtrack during graph traversal. Overall, the reviewers lean toward accepting the work, following the clarifications provided during the rebuttal period.

Strengths
- Explicit Action Space Formulation (Reviewers 8bJc, fbd6, vsHb): The formalization of graph reasoning into four atomic operations provides a traceable framework. The Backjump can address the common dead-end problem where agents get trapped in unproductive reasoning paths.
- Effective Training Pipeline (Reviewers fbd6, 8bJc, tkwC): The two-stage training strategy enables basic navigation and complex self-correction capabilities.

Weaknesses
- Computational Training Overhead (Reviewer fbd6): The reliance on an LLM-based verifier for step-wise faithfulness rewards during the RL phase introduces significant GPU-hour costs. While asynchronous evaluation helps, the scalability of this training paradigm for extremely large-scale applications remains a concern.

- Incremental Conceptual Novelty (Reviewer fbd6, vsHb): From a theoretical perspective, the integration of backtracking into an RL framework is not a fundamental paradigm shift, as backtracking is a well-established concept in classical search and planning.